# Lattice-mismatch-free construction of III-V/ chalcogenide core-shell heterostructure nanowires

Fengjing Liu[1], Xinming Zhuang[1], Mingxu Wang[1], Dongqing Qi[2], Shengpan Dong[3], SenPo Yip[4], Yanxue Yin[1], Jie Zhang[1], Zixu Sa[1], Kepeng Song ®[2] ✉, Longbing He ®[3], Yang Tan[1], You Meng[5], Johnny C. Ho ®[4,5] ✉, Lei Liao ®[6], Feng Chen ®[1] & Zai-xing Yang ®[1] ✉

Growing high-quality core-shell heterostructure nanowires is still challenging due to the lattice mismatch issue at the radial interface. Herein, a versatile strategy is exploited for the lattice-mismatch-free construction of III-V/chalcogenide core-shell heterostructure nanowires by simply utilizing the surfactant and amorphous natures of chalcogenide semiconductors. Specifically, a variety of III-V/chalcogenide core-shell heterostructure nanowires are successfully constructed with controlled shell thicknesses, compositions, and smooth surfaces. Due to the conformal properties of obtained heterostructure nanowires, the wavelength-dependent bi-directional photoresponse and visible light-assisted infrared photodetection are realized in the type-I GaSb/GeS core-shell heterostructure nanowires. Also, the enhanced infrared photodetection is found in the type-II InGaAs/GeS core-shell heterostructure nanowires compared with the pristine InGaAs nanowires, in which both responsivity and detectivity are improved by more than 2 orders of magnitude. Evidently, this work paves the way for the lattice-mismatch-free construction of core-shell heterostructure nanowires by chemical vapor deposition for next-generation high-performance nanowire optoelectronics.

Due to their unique quasi-one-dimensional geometry and intriguing physical properties, III–V nanowires (NWs) have attracted enormous attention in electronics, optoelectronics, quantum computing, etc[1–8]. However, the increasingly versatile and high-end application demands have significantly complicated the design of functional devices, which is sometimes difficult to realize solely based on the intrinsic physical and semiconductor properties[9]. In this case, core–shell heterostructure NWs have been extensively explored to tackle these needs[10–12].

Particularly, in core–shell heterostructure NWs, their electronic and optoelectronic properties can be effectively modulated by combining the advantages of constituent semiconductors and rational design of radial band alignment owing to the large interfacial area[13]. For instance, Pendharkar et al. demonstrated parity-preserving and magnetic field-resilient superconductivity in InSb NWs with Sn shells by constructing semiconductor-superconductor heterostructures[12]. Moreover, Jagadish and his team demonstrated the type-I band alignment of GaAs/InGaAs

[1]School of Physics, State Key Laboratory of Crystal Materials, Shandong University, 250100 Jinan, China. [2]School of Chemistry and Chemical Engineering, Shandong University, 250100 Jinan, China. [3]SEU-FEI Nano-Pico Center, Key Lab of MEMS of Ministry of Education, Collaborative Innovation Center for Micro/Nano Fabrication, Device and System, Southeast University, 210096 Nanjing, China. [4]Institute for Materials Chemistry and Engineering, Kyushu University, 816-8580 Fukuoka, Japan. [5]Department of Materials Science and Engineering, City University of Hong Kong, 999077 Hong Kong, China. [6]Key Laboratory for Micro-Nano Optoelectronic Devices of Ministry of Education, School of Physics and Electronics, Hunan University, 410082 Changsha, China. ✉e-mail: kpsong@sdu.edu.cn; johnnyho@cityu.edu.hk; zaixyang@sdu.edu.cn

core–shell NWs with great potential for lasers[7]. For photodetectors and photovoltaics, type-II band alignment core–shell NWs are advantageous due to their efficient spatial separation of the photogenerated carriers[14,15]. In this regard, different types of band alignment can be tailored and employed for different applications[13].

Despite these recent developments, solving the issue of lattice mismatch for constructing and utilizing high-quality core–shell heterostructure NWs is still challenging[16,17]. At present, it is usually adopted to grow a very thin shell layer or select the constituent materials with very close lattice constants to minimize the adverse impact of lattice mismatch[18–20]. In fact, there would not be any lattice mismatch issue when the amorphous shell, rather than the crystalline shell, is employed. At the same time, chalcogenide semiconductors, composed of chalcogenide (S, Se, or Te) and Ge (As, Sb, Ga, or P) atoms via covalent-bond networks, are reported as amorphous semiconductors, displaying competitive applications in phase-change memories, photodetectors, solar cells, image sensors, etc[21–23]. More importantly, it has been revealed that chalcogenide atoms were the preferred surface passivation agents for III–V semiconductors[24–26], resulting in abundant chalcogenide atoms on the surfaces. Especially for Sb-based III–V NWs growth, the chalcogenide atoms have also been utilized as surfactants to stabilize the sidewalls and thus minimize unintentional radial growth[27]. Obviously, the abundant chalcogenide atoms on the surfaces would facilitate conformal growth of the amorphous chalcogenide shell around the III–V NW core, which potentially leads to the effective lattice-mismatch-free construction of core–shell heterostructure NWs.

In this work, the narrow bandgap III–V and amorphous chalcogenide semiconductors are adopted to design and demonstrate the lattice-mismatch-free construction of core–shell heterostructure NWs by chemical vapor deposition (CVD). Benefiting from the surfactant function of chalcogenide atoms, the amorphous chalcogenide shells can grow conformally around the III–V NWs, resulting in the anticipated GaSb/GeS, GaAs/GeS, InGaAs/GeS, and GaSb/GeSe core–shell heterostructure NWs. The diameter and shell thickness of as-constructed core–shell NWs can be simply controlled by the growth duration. Through the rational design of shell composition and thickness, the core–shell NWs exhibit controllable band alignments and intriguing photodetection behaviors. Specifically, the GaSb/GeS core–shell NW exhibits the type-I heterostructure with unique wavelength-dependent bi-directional photoresponse behavior. These properties give excellent visible light-assisted infrared photodetection performance, which is attributable to hot-carrier trapping states in the amorphous GeS shells. Compared to the pristine InGaAs NW, the InGaAs/GeS core–shell NW displays the type-II heterostructure, revealing enhanced photodetection performance with responsivity ($R$) and detectivity ($D^*$) increased by about 2 orders of magnitude. Furthermore, based on these high-quality InGaAs/GeS core–shell heterostructure NWs, the as-fabricated infrared image sensor delivers a good infrared imaging ability. All these findings can evidently demonstrate a simple strategy for the lattice-mismatch-free construction of III–V/chalcogenide core–shell heterostructure NWs, which paves the way for further regulating and improving the device performance of semiconductors NWs.

## Results

### Lattice-mismatch-free construction of GaSb/GeS core–shell heterostructure NWs

GaSb NWs, as a typical narrow bandgap p-type semiconductor, possess the highest hole mobility among all III–V semiconductors, attracting enormous attention in high-speed electronic and optoelectronic devices[28–30]. Here, the amorphous chalcogenide of GeS is first explored for the conformal growth around GaSb NWs by a two-step synthetic methodology, involving (i) the growth of GaSb NWs via sulfur-assisted CVD followed by (ii) the growth of GeS shells. As shown in Fig. S1, the GaSb/GeS core–shell heterostructure NWs can be simply prepared by independently controlling the heating duration and sequence of the

heating zones in a three-temperature-zone tube furnace (see "Methods" for details). From the transmission electron microscope (TEM) image (Fig. 1a), the as-prepared NW shows apparent contrast between the core and shell. It is also obvious that the shell layer conformally wraps around the core, indicating the core–shell heterostructure NW being successfully constructed. The microstructure of as-constructed core–shell heterostructure NWs is then evaluated by cross-sectional high-angle annular dark field scanning transmission electron microscopy (HAADF STEM), as presented in Fig. 1b, c. It is further confirmed that an amorphous shell covers uniformly around the GaSb core. Meanwhile, there are continuous lattice fringes in the GaSb core region and clear reciprocal lattice spots extracted by fast Fourier transform (FFT), indicating the GaSb core is single crystallinity with a <111> growth direction, where this direction is consistent with the previous reports[31]. The homogeneity of the core–shell heterostructure NW is also excellent, as depicted in Fig. S2. For checking the possibility of axial growth, the tails of as-constructed GaSb/GeS core–shell heterostructure NWs are observed by TEM. As shown in Fig. S3, all the NWs exhibit conformal core–shell heterostructure. No axial growth of amorphous GeS is observed.

To further confirm the compositions of the core and shell layers, energy-dispersive X-ray spectroscopy (EDS) elemental mapping is acquired under STEM mode. As depicted in Fig. 1d–g, it is witnessed that the Ga and Sb elements are dominated in the core, while the Ge and S elements are distributed in the shell. Owing to the Ga$^+$ ion being used in the preparation of cross-sectional TEM specimens, Ga exists in the entire NW. It is worth noting that the S elements distribute evenly in the shell, whereas the content of Ge elements increases gradually along the radial direction. The non-uniform ratio of Ge and S elements in the shell indicates the construction of a disordered network, which verifies the formation of an amorphous shell[32,33]. In addition, the diameter and shell thickness of the GaSb/GeS core–shell heterostructure NWs can be regulated reliably by tuning the growth duration of GeS (Fig. S4). As revealed in Fig. 1h, i, when the growth duration of GeS is 10 s, 30 s, and 60 s, the diameter of as-constructed core–shell NWs is modulated from $52.0 \pm 8.4$ nm to $58.2 \pm 8.6$ nm and $62.1 \pm 8.9$ nm, along with the shell thickness increases from $11.3 \pm 2.0$ nm to $14.7 \pm 1.9$ nm and $19.5 \pm 4.7$ nm, respectively. Based on the X-ray diffraction (XRD) patterns as depicted in Fig. 1j, there are only the diffraction peaks of zinc blende GaSb[34], which further confirms the shell of GeS being amorphous. In short, all results demonstrate that the simple CVD method can successfully construct the GaSb/GeS core–shell heterostructure NWs with controllable diameter and shell thickness.

### Construction mechanism of GaSb/GeS core–shell heterostructure NWs

To shed light on the construction mechanism of GaSb/GeS core–shell heterostructure NWs, the X-ray photoelectron spectroscopy (XPS) spectra of typical Ga 3$d$, S 2$p$, Ge 3$d$, and Sb 4$d$ are selected for analysis in Fig. 2, while all peaks are calibrated with the C 1$s$ peak (284.80 eV) and fitted through Gauss–Lorentz fitting. For pristine GaSb NWs, typical Ga-Sb (18.93 eV), S-Sb (2$p_{3/2}$ at 161.68 eV, 2$p_{1/2}$ at 163.38 eV), Sb-Ga (4$d_{5/2}$ at 31.58 eV, 4$d_{3/2}$ at 32.83 eV) and Sb-S (34.36 eV) bonds can be identified, indicating the S surfactants are utilized successfully to stabilize the NW sidewalls, and thus minimize the unintentional radial growth[27]. On the other hand, there are peaks of S-Ge (2$p_{3/2}$ at 161.98 eV, 2$p_{1/2}$ at 163.18 eV) and Ge-S (30.69 eV) but without the peaks of Ga-Sb or Sb-Ga witnessed in the GaSb/GeS core–shell heterostructure NWs before Ar$^+$ ion etching, which suggests the shell being GeS. The peaks of Ga-Sb, Sb-S and Ge-S appear at the etching time of 10 s. These peaks become stronger once the etching time reaches 40 s. Furthermore, the binding energies of both S-Ge and Ge-S bonds appear to have a blue shift with etching, indicating that the relative content of Ge decreases from the surface to the inner shell. This observation is consistent with the EDS elemental mapping result and previous literatures[35,36]. All

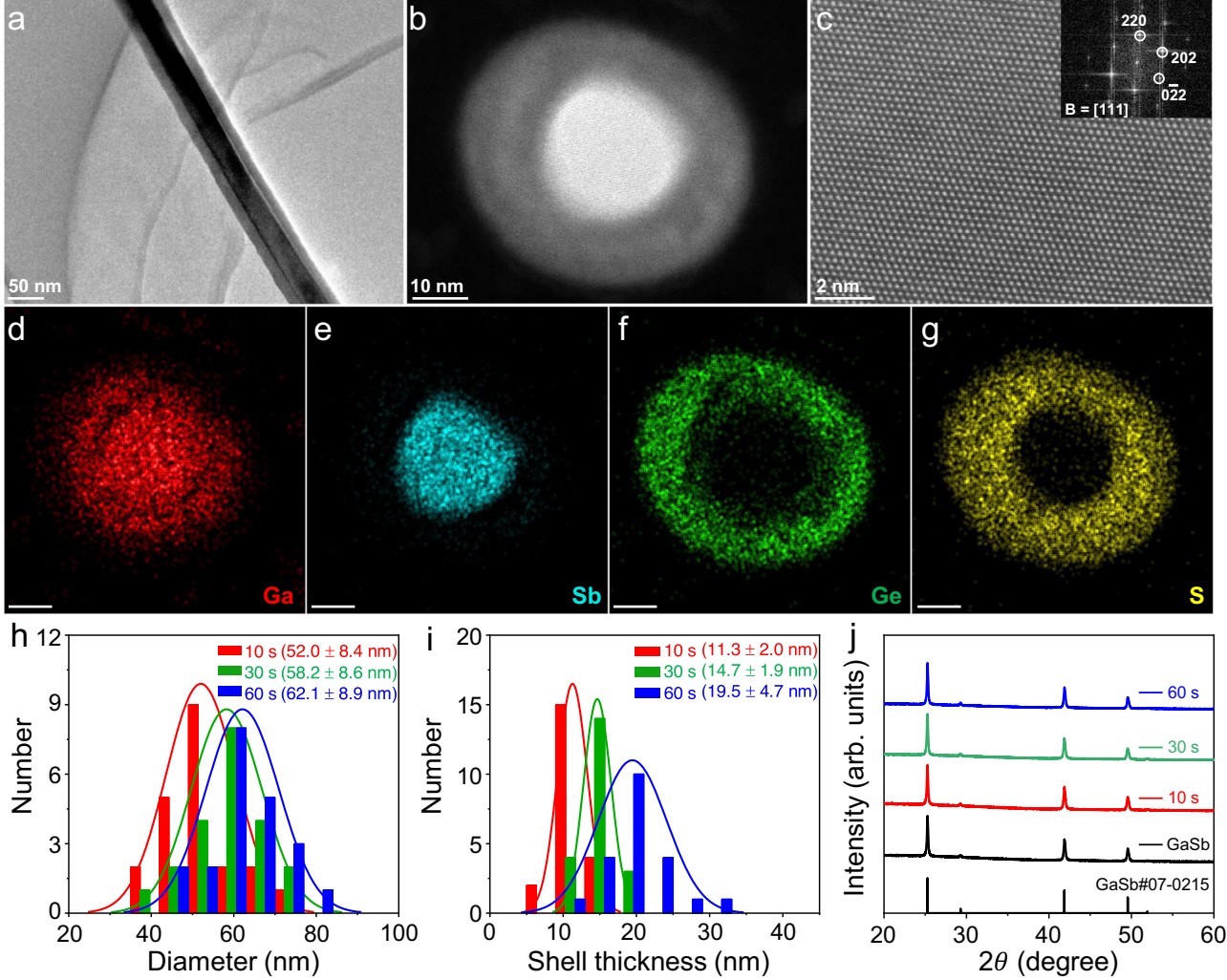

**Fig. 1 | Lattice-mismatch-free construction of core−shell heterostructure NWs, taking GaSb/GeS as an example. a–c** TEM, cross-sectional HAADF STEM, and partial enlargement HAADF STEM images of the as-constructed core−shell heterostructure NWs. The inset of (**c**) shows the corresponding FFT pattern. **d–g** EDS elemental mapping images of Ga, Sb, Ge, and S, respectively. All the scale bars are 10 nm. **h–j** Diameter statistic, shell thickness statistic, and XRD patterns of as-constructed core−shell heterostructure NWs with different growth duration.

findings verify the successful construction of GaSb/GeS core−shell heterostructure NWs.

Moreover, based on the results of XPS and STEM, an illustrative atomic model is constructed to present the schematic details of GaSb/ GeS core−shell heterostructure NWs as shown in Fig. 2d. Specifically, as a surfactant during the NW growth process, S atoms evenly distribute on the surface of GaSb NWs[27], which facilitates the subsequent radial conformal growth of the amorphous GeS shell[37]. In fact, other chalcogens, such as Se and Te, also tend to bond with the surface atoms of III−V semiconductors[24], and will also participate as the intermediary role in the conformal growth of shell layers. On the other hand, the amorphous characteristics of chalcogen semiconductors (such as GeS, GeSe, etc.) can effectively release the strain caused by lattice mismatch with the underlying materials. These features would contribute to the lattice-mismatch-free construction of III−V/chalcogenide core−shell heterostructure NWs. In short, the developed strategy is simple and versatile, which can also be easily extended to the lattice-mismatch-free construction of other material systems of core−shell heterostructure NWs.

**The versatility of lattice-mismatch-free construction of III−V/ chalcogenide core−shell heterostructure NWs**

After understanding the construction mechanism, other typical narrow-bandgap III−V NWs, including binary GaAs and ternary InGaAs,

are selected as the core, while other amorphous chalcogenide semiconductors, such as GeSe, are chosen as the shell to demonstrate further the versatility of the lattice-mismatch-free construction strategy for III−V/chalcogenide core−shell heterostructure NWs. As shown in the scanning electron microscope (SEM) images in Fig. 3a, d, g, all samples show the uniform morphology of NWs with long lengths and smooth surfaces. Based on the high-resolution TEM (HRTEM) images (Fig. 3b, e, h), all NWs are core-shell nanostructures, along with the clear contrasts between the cores and shells. The shells look inhomogeneous around the NW cores in the HRTEM images of Fig. 3b, e, h. However, in the cross-sectional HAADF STEM images of Figs. 1 and S2, low-resolution STEM images of Fig. 3c, f, i and TEM images of Figs. S3, the shells are homogeneous around the NW cores. The inhomogeneity may be caused by the shooting angle during the observation of HRTEM. Under the smaller view, a slight incline of the sample will cause the observation of inhomogeneity. The continuous lattice fringes and clear reciprocal lattice spots (extracted by FFT) are observed in the core region, while there are no lattice fringes and reciprocal lattice spots in the shell layer, indicating the excellent crystallinity of the core and the amorphous characteristic of the shell. Importantly, the core−shell integration of the above semiconductors can realize a variety of heterostructures, as shown in Fig. S5. They include the type-I heterostructures (e.g., GaSb/GeS) and the type-II heterostructures

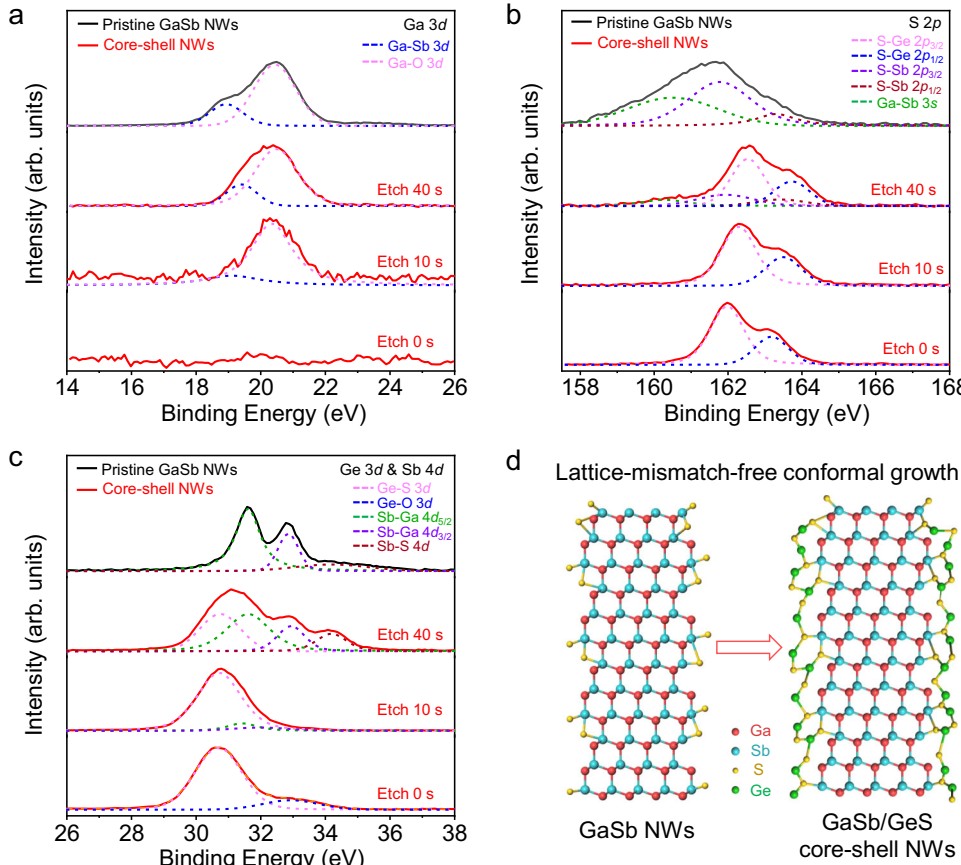

**Fig. 2 | Surface states and construction schematics of GaSb/GeS core–shell heterostructure NWs. a–c** XPS spectra of Ga 3d, S 2p, Ge 3d, and Sb 4d of pristine GaSb NWs (grown by surfactant-assisted CVD method) and GaSb/GeS core–shell heterostructure NWs, respectively. **d** Construction schematics of GaSb/GeS core–shell heterostructure NWs.

(e.g., GaAs/GeS, InGaAs/GeS, and GaSb/GeSe), which are promising for building various high-performance optoelectronics devices[13].

The compositions are then evaluated by EDS elemental mappings as presented in Fig. 3c, f, i. Obviously, all the elements of III–V semi-conductors are distributed uniformly in the core, whereas the elements of chalcogen semiconductors are homogenously distributed along the entire NW, indicating the success in the lattice-mismatch-free construction of III–V/chalcogenide core–shell heterostructure NWs by the simple CVD method. In fact, owing to the sulfur surfactant used in the NWs growth process, the unsaturated surface-terminated Sb is effectively stabilized to form the stable S-Sb bond, which results in a little larger distribution diameter of Ga than Sb[27,34,38], as shown in Fig. 3i. Moreover, the diameter and the shell thickness of the as-constructed core–shell heterostructure NWs can be regulated simply by the growth times, as shown in Fig. S6. The crystallinity of all as-constructed core–shell heterostructure NWs is also assessed by XRD, as shown in Fig. S6. The diffraction peaks of III–V semiconductors are observed only, further confirming the success in the lattice-mismatch-free construction of III–V/chalcogenide core–shell heterostructure NWs. In brief, all results demonstrate the developed strategy is an efficient and versatile approach for the lattice-mismatch-free con-struction of core–shell heterostructure NWs.

**Photodetection behaviors of the lattice-mismatch-free con-structed III–V/chalcogenide core–shell heterostructure NWs**

Owing to the intriguing band alignment, the effective construction of these semiconductor heterostructures enables many advanced optoelectronics devices[39–41]. Figure 4 shows the photodetection characteristics of as-constructed type-I GaSb/GeS core–shell

heterostructure NWs by adopting the metal-semiconductor-metal (MSM) photodetector configuration. As shown in the I-V curves of Fig. S7, with the increase of shell thickness, the $I_{DS}$ of as-studied MSM photodetectors decreases, indicating the GaSb cores act as the main conductive channels, and the GeS shells limit the currents owing to the formation of heterojunctions at the contacts. It is worth mentioning that in addition to the classical thermionic emission, tunneling or defect-assisted tunneling also plays role on the holes injection and collection processes[42,43]. With a thin shell, the barrier is possibly thin and low, benefiting to the holes tunneling. As a result, with the increase of shell thickness, the $I_{DS}$ decreases. Because both the holes injection and collection processes pass across the GeS shell, it also plays important role on the transport of photogenerated carriers at the contacts, and the details will be discussed later. As shown in Fig. 4a, the broad-spectrum photodetection behaviors of GaSb/GeS core–shell heterostructure NWs are dependent on the shell thickness and wave-length. With a thin shell thickness of $11.3 \pm 2.0$ nm, the GaSb/GeS core–shell heterostructure NW displays a positive photoresponse, similar to the pristine GaSb NW (Fig. S8) but with weaker photo-currents. Once the shell thickness is increased larger than $14.7 \pm 1.9$ nm, the GaSb/GeS core–shell heterostructure NW exhibits unusual wavelength-dependent bi-directional photodetection behaviors, which display a negative photoresponse in the wavelength of 405–785 nm and a positive photoresponse in the wavelength of 850–1550 nm.

With a thicker shell, the photocurrent becomes weaker. In this case, the photodetection behaviors of core–shell heterostructure NWs with a medium shell thickness are further studied, as presented in Fig. 4b-f. With the increasing laser intensity, the negative

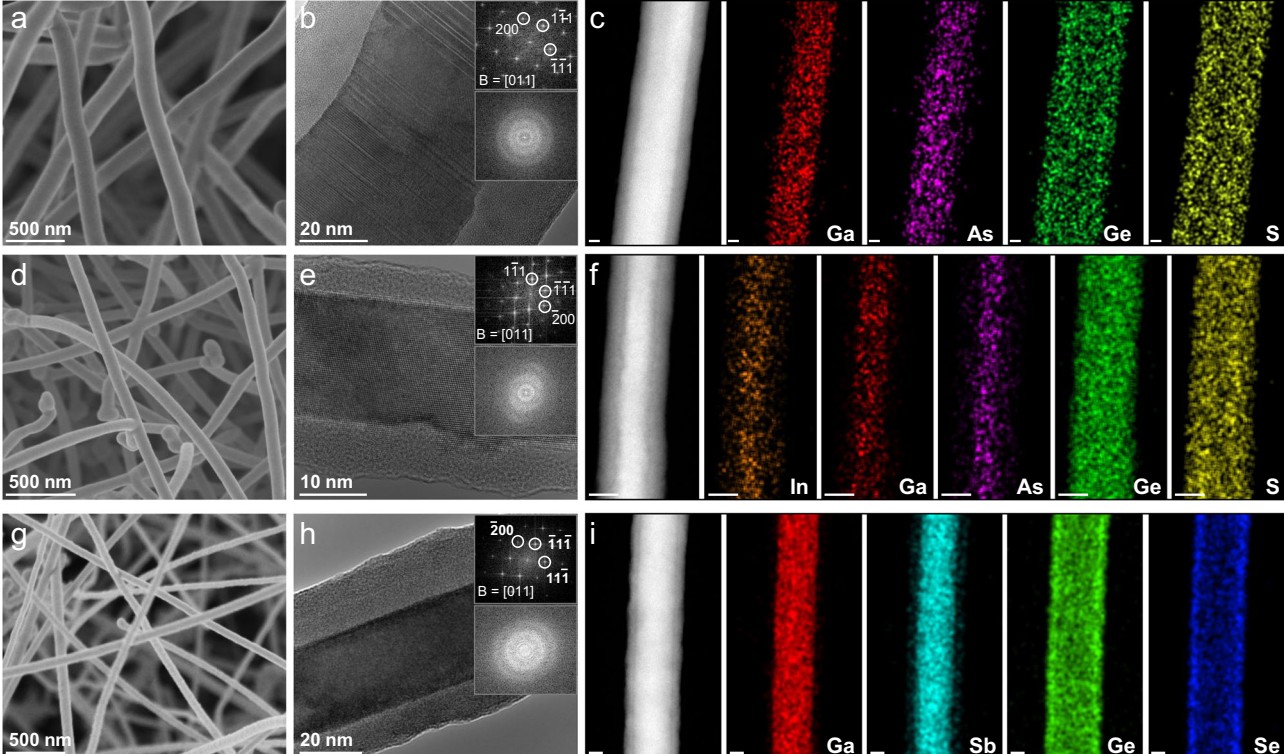

**Fig. 3 | The versatility of lattice-mismatch-free construction of III–V/chalcogenide core–shell heterostructure NWs. a–c** The SEM image, HRTEM image and EDS elemental mapping images of GaAs/GeS core–shell heterostructure NWs. **d–f** The SEM image, HRTEM image and EDS elemental mapping images of InGaAs/ GeS core–shell heterostructure NWs. **g–i** The SEM image, HRTEM image and EDS elemental mapping images of GaSb/GeSe core–shell heterostructure NWs. The insets in (**b**, **e**, **f**) show the corresponding FFT images of the core and shell regions, respectively. All the scale bars in (**c**, **f**, **i**) are 20 nm.

photocurrent decreases under the illumination of a 405 nm laser. The positive photocurrent increases under the irradiation of a 1310 nm laser. Interestingly, the photocurrent changes from negative to positive under the illumination of a 785 nm laser. Furthermore, utilizing the bi-directional photodetection feature, the visible light-assisted improved infrared photodetector performance is observed in these core–shell heterostructure NW photodetectors, as illustrated in Fig. 4e, f. With the assistance of a 405 nm laser, the dark current of the infrared photodetector is significantly suppressed from 256 to 165 nA. At the same time, the photocurrent increases from 26 nA to 42 nA. This visible light-assisted behavior is also observed under the illuminations of 520 nm and 635 nm lasers, as shown in Fig. S9. More importantly, the rise/decay time ($t_r/t_d$), defined as the time for the photocurrent to increase/decrease from 10/90% to 90/10%, is significantly shortened from 72.0 ms to 8.0 ms and 2.0 s to 12.0 ms, respectively[44]. Also, both the $R$ and $D^*$ are improved by several folds, namely, the values up to 400 A W$^{-1}$ and $9.8 \times 10^{10}$ Jones, respectively, as shown in Fig. S10. Additionally, as shown in Fig. S11, with an ultra-thin shell of 2 nm (growth time is 1 s), the GaSb/GeS core–shell heterostructure NW exhibits larger photocurrents compared to the pristine GaSb NW, which is contributed to the surface passivation effect of the ultra-thin GeS shell. At the same time, due to the negative photoresponse caused by the GeS shell, a reduced photocurrent is also observed at the near ultraviolet waveband of 405 nm. This surface passivation effect is also observed in GaSb NWs with the epitaxial shells of larger bandgap Al$_2$O$_3$, as shown in Fig. S12. The results show that the epitaxial larger bandgap Al$_2$O$_3$ shells can passivate the surface charge trappings of GaSb NWs effectively. Compared to the as-constructed III–V/chalcogenide core–shell heterostructure NWs, wavelength-dependent bi-directional photodetection behavior, visible light-assisted infrared photodetection behavior, and faster response times are not observed in the GaSb/Al$_2$O$_3$ core–shell heterostructure NWs. Furthermore,

benefiting from the as-constructed core–shell nanostructure, the as-fabricated photodetector exhibits stable wavelength-dependent bi-directional photodetection and visible light-assisted infrared photodetection behaviors after being stored in the atmospheric environment for 30 days, as shown in Fig. S13.

In order to fully understand the bi-directional mechanism and visible light-assisted behaviors, a band structure model is proposed and depicted in Fig. 4g. With the bandgaps of 0.70 eV and 1.50 eV for GaSb core and GeS shell, both of core and shell can absorb the visible light (<827 nm) and generate electron-hole pairs (processes I and I'). In the case of the amorphous GeS shell, the photogenerated hot holes will be trapped by the trapping states (process II), which is similar to the previous works' phenomenon[36,45,46]. Meanwhile, the photogenerated electrons would transfer to the GaSb core, which is driven by the built-in electric field of the as-constructed heterostructure NW (namely process II'). As a result, the injected electrons are going to recombine with the holes in the GaSb core (namely process III), resulting in the decreased hole concentration. On the other hand, the photogenerated electron-hole pairs in the GaSb core would increase the hole concentration. In fact, the hole concentration of GaSb core affect seriously the hole transport process at the contact[47], as illustrated in Fig. S14. In a word, the decreased hole concentration in GaSb core leads to the increase of the hole transport barrier, leading to a decreased current (negative photoresponse). On the contrary, the increased hole concentration in GaSb core leads to the decrease of the hole transport barrier, leading to an increased current (positive photoresponse). Additionally, the thickness of the GeS shell also has an important impact on the hole transport in photorespose behaviors, owing to the tunneling. The details can be found in the supplementary information. Obviously, the negative and positive photoresponse coexist and compete in the as-constructed core–shell heterostructure NW under the illumination of visible light, resulting in the interesting shell thickness dependent bi-directional

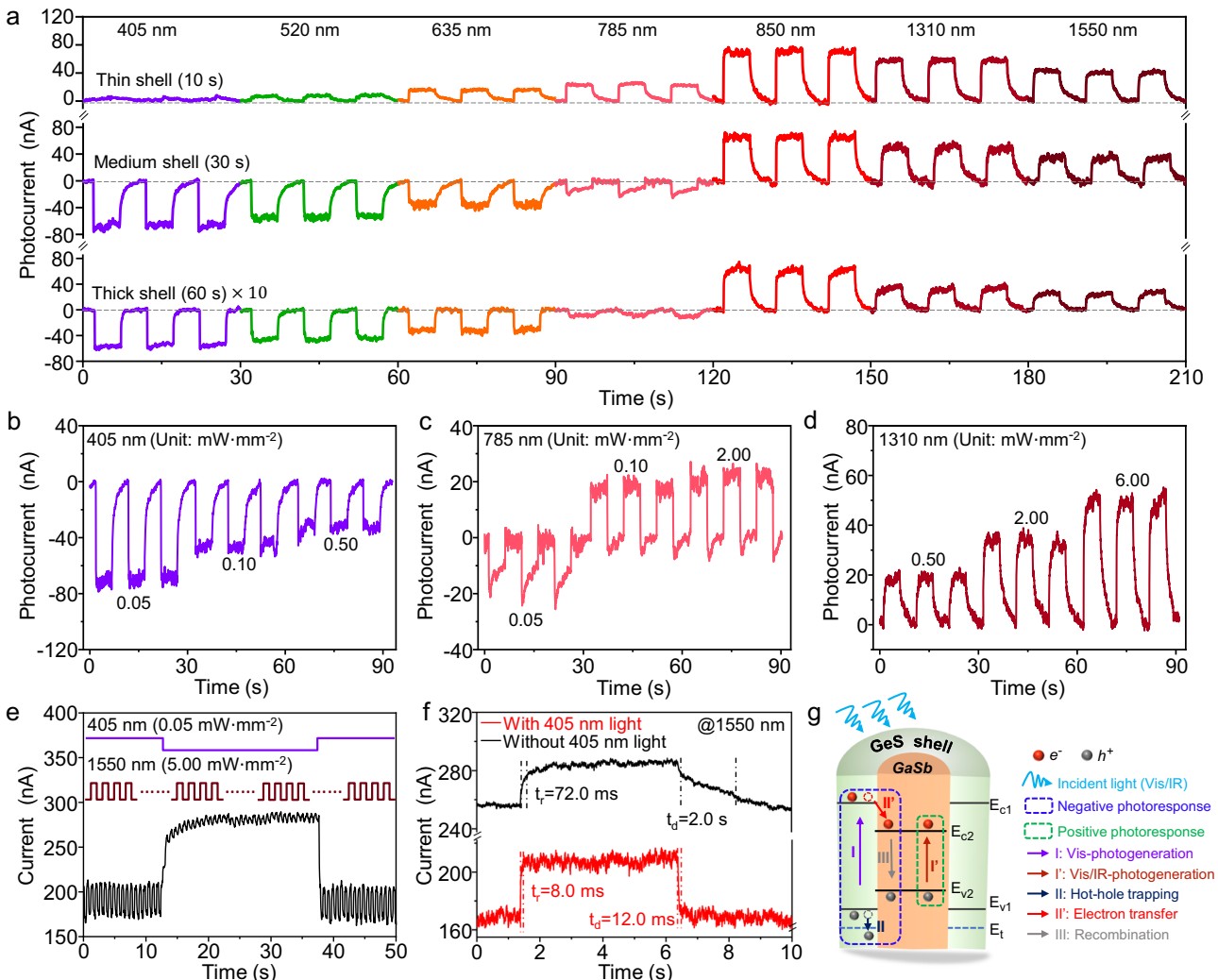

**Fig. 4 | Photodetection behavior of type-I GaSb/GeS core–shell heterostructure NWs. a** Wavelength-dependent broad-spectrum photodetection performances at 1 V bias. The laser intensities from 405 nm to 785 nm and from 850 nm to 1550 nm are 0.05 mW mm⁻² and 6.00 mW mm⁻², respectively. **b–d** I-t curves of as-fabricated photodetector operated under the illuminations of 405, 785, and 1310 nm lasers with different intensities, respectively. **e** Visible light-assisted infrared photodetection performance. **f** The temporal photoresponse characteristics of 1550 nm laser (5.00 mW mm⁻²) with and without the assist of 405 nm laser. **g** Schematic for type-I GaSb/GeS core–shell heterostructure NWs photodetectors.

photodetection behavior. With a thin shell, the proportion of visible light absorbed by GeS is limited, and the visible light is mainly absorbed by GaSb, resulting in a positive photoresponse. With a thick shell, the proportion of visible light absorbed by GeS increases, while that absorbed by GaSb decreases, resulting in a negative photoresponse. When the core–shell heterostructure NWs are exposed to infrared light (>850 nm), only GaSb core can absorb the irradiation and generate the electron-hole pairs (process I′), resulting in the positive photodetection behavior in the infrared waveband. Furthermore, with the assistance of visible light, the dark current of the photodetector for infrared light would be reduced due to the photogenerated electrons in the GeS shell recombining with the free holes in the GaSb core (Fig. 4e). Meanwhile, the trapped photogenerated holes in the GeS shell leads to the photo-gating effect, which would passivate the interface defects (field-effect passivation) and prolong the lifetime of infrared light excited electron-hole pairs in the GaSb core, ultimately leading to the improvement of infrared photodetection performances[48–50]. It is worth pointing out that the shell thickness also plays an important role in the visible light-assisted behavior, as shown in Fig. S15. In a word, a careful design of appropriate shell thickness is necessary for high-performance bi-directional photodetection and visible light-assisted photodetection.

Apart from building the type-I heterostructures, we can also construct the type-II core–shell heterostructure NWs (such as InGaAs/GeS) using our developed synthetic strategy and evaluate their photodetection characteristics. As compiled in Fig. 5a, it is found that both NWs exhibit broad-spectrum photodetection in the wavelength range between 405 nm and 850 nm. Compared with the pristine InGaAs NW, the photocurrent of InGaAs/GeS core–shell heterostructure NW is larger by more than 3 orders of magnitude in the visible waveband and more than 2 orders in the near-infrared waveband. Also, both the $R$ and $D^*$ are improved by more than 2 orders of magnitude (Fig. 5b), with values up to 61.2 A W⁻¹ and 6.8 × 10¹¹ Jones, respectively, under the illumination of an 850 nm laser. Moreover, the response time is shortened by more than 50 times, that is, the rise time is only 3.3 ms and the decay time is only 2.9 ms for the InGaAs/GeS core–shell heterostructure NW (Fig. 5c). The significant improvement in the photodetection performance is attributed to the successful construction of the type-II p-n core–shell heterostructure[51], as discussed in Fig. S16. The good switching on/off repeatability of the core–shell heterostructure NW photodetector is also demonstrated in Fig. 5d. Benefiting from the excellent infrared photodetection performance, a 5 × 5-pixel infrared image sensor based on ordered InGaAs/GeS core–shell

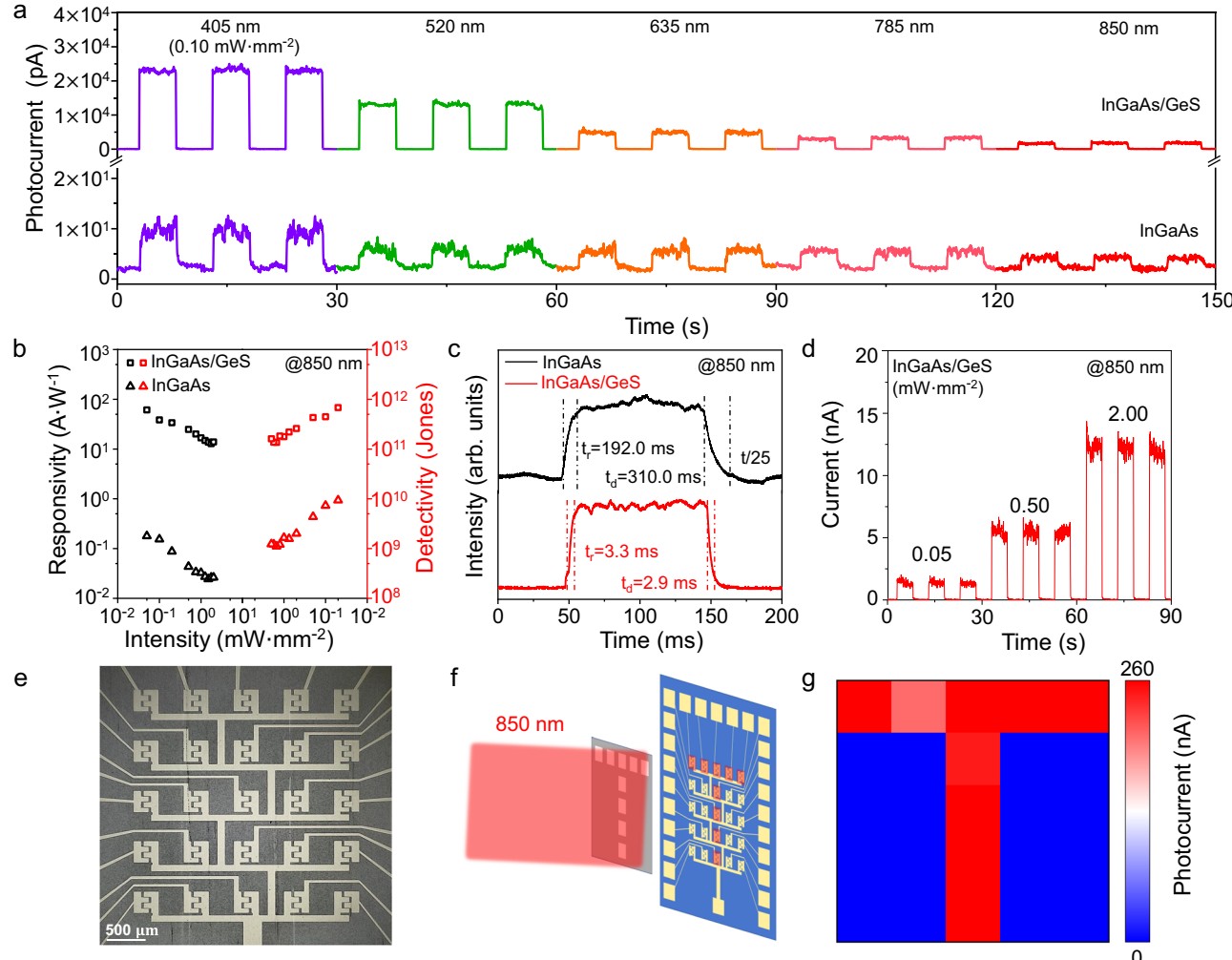

**Fig. 5 | Photodetection behavior of type-II InGaAs/GeS core–shell heterostructure NWs. a** Broad-spectrum photodetection performances at 1 V bias. **b** $R$ and $D^*$ of as-fabricated InGaAs and InGaAs/GeS core–shell heterostructure NWs MSM photodetectors under the illumination of an 850 nm laser. **c** The temporal photoresponse characteristics. The time axis of the InGaAs NW is shrunken by a factor of 25. **d** Switching on/off repeatability test of the core–shell heterostructure NW photodetector. **e**–**g** Infrared photodetection imaging demonstration by the ordered InGaAs/GeS core–shell heterostructure NWs arrays.

heterostructure NWs is prepared and illustrated in Fig. 5e. The ordered InGaAs/GeS core–shell NWs array is realized by a well-developed contact printing technology[52], exhibiting excellent infrared detection capability with photocurrent up to 150 nA under the illumination of an 850 nm laser (Fig. S17). The corresponding imaging performances is presented in Fig. 5f, g. A hollow target mask of letter "T" is placed in front of the photodetector array. It is witnessed that the detected information of the target is well demonstrated by the image sensor array.

Up to now, many efforts have been focused on constructing III–V core–shell heterostructure NWs, but the study on their photodetection behaviors is still rare, as shown in Table S1[53–57]. On the contrary, by adopting the developed lattice-mismatch-free construction approach, a variety of III–V/chalcogenide core–shell heterostructure NWs with desired band alignment can be constructed, along with intriguing photodetection behaviors. Furthermore, as shown in Fig. S18, the lattice-mismatch-free construction approach can also be used to grow chalcogenides core–shell heterostructure NWs. Owing to the rational band alignment, the as-constructed CdS/GeS core–shell heterostructure NWs exhibit excellent photodetection performance with extended spectrum detection range and much larger photocurrent compared to that of pristine CdS NWs. In short, the lattice-mismatch-free construction of core–shell heterostructure NWs with rational band alignment is an effective strategy to realize next-generation high-performance optoelectronics devices.

## Discussion

In conclusion, a versatility strategy is demonstrated successfully for the lattice-mismatch-free construction of III–V/chalcogenide core–shell heterostructure NWs, including GaSb/GeS, GaAs/GeS, InGaAs/GeS, and GaSb/GeSe NWs. All the shells are amorphous chalcogenide semiconductors with controlled thicknesses, compositions, and smooth surfaces, where they grow conformally from the chalcogenide surfactant atoms around III–V NWs by the simple CVD method. With the rational design of the band alignment, photodetection behaviors of different III–V/chalcogenide core–shell heterostructure NWs are effectively modulated. All these results can evidently provide a strategy for constructing lattice-mismatch-free core–shell heterostructure NWs by a simple method for next-generation high-performance optoelectronics.

## Methods
### Conformal growth of core–shell heterostructure NWs
The core–shell heterostructure NWs are constructed by a two-step synthetic methodology in a three-temperature-zone tube furnace. In general, chalcogenide semiconductors powders (such as GeS or GeSe)

are placed in upstream zone 1, high purity III–V source powders (such as GaSb, GaAs, InAs or their mixture powders) are placed in zone 2, and the growth substrate with metal catalyst is placed in the downstream zone 3. Firstly, III–V NWs are prepared in the last two zones (zone 2 and zone 3). After the growth of III–V NWs, zone 1 is heated to the growth temperature of chalcogenide semiconductors immediately, and the holding time is adjusted to realize the controllable growth of the amorphous shell. It is worth pointing out that zone 2 needs to cool naturally to the same temperature as zone 1. After reaching the set time, the three zones would stop heating together, and the furnace chamber would be opened for rapid cooling to room temperature. The growth details of core–shell heterostructure NWs can be found in Table S2.

## Material characterization

The morphologies of the as-constructed core–shell NWs are characterized by SEM (Helios G4 UC, Thermo Scientific) and TEM (JEM-F200, JEOL). Cross-sectional TEM specimens are prepared using a focused ion beam system (Helios G4 UC, Thermo Scientific). HRTEM and HAADF-STEM associated with EDS elemental mapping are performed at 300 kV on an aberration-corrected transmission electron microscope equipped with a Super-X EDS detector (Spectra 300, Thermofisher). The crystal phase and phase purity of as-prepared NWs are verified by XRD (D8 Advance, Bruker). The chemical composition and elemental valence are analyzed by XPS (Nexsa, Thermo Scientific). For thickness-dependent XPS spectra, the GaSb/GeS core–shell NWs are thinned by $Ar^+$ ion etching, which is attached to the XPS equipment. The band structures of the as-constructed core–shell heterostructure NWs are determined by the UPS (ESCALAB XI+, ThermoFisher).

## Devices fabrication

The fabrication of individual single NW photodetector and arrays photodetector is as follows. For a single NW photodetector, the as-prepared NWs are suspended in ethanol solution by ultrasonication and then transferred onto Si substrates with a 50 nm thick $SiO_2$ by drop-casting. And for arrays photodetector, ordered core–shell NWs are transferred onto the Si substrates with a 50 nm thick $SiO_2$ by contact printing craft. The electrodes are patterned by UV photolithography, and a 50 nm thick Ni film is thermally deposited, followed by a standard lift-off process. The hollow mask (Cr/Ni 5/50 nm) is fabricated on a quartz glass by UV photolithography, thermal evaporation, and standard lift-off process.

## Electronic and optoelectronic characterization

Both the electrical and photodetection performance of as-constructed NWs photodetectors are characterized by an Agilent B1500A semiconductor analyzer connected to a probe station at room temperature. The diode lasers serve as light sources. The periodic optical signal is generated by the waveform generator controlling the power input of the diode laser. The laser is directly illuminated on the photodetector through an optical fiber without focusing. The laser intensity is calibrated by a power meter.

## Data availability

Relevant data supporting the key findings of this study are available within the article, the Supplementary Information file, and the Source data file. All raw data generated during the current study are available from the corresponding authors upon request. Source data are provided with this paper.

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

## Acknowledgements

We acknowledge the National Key R&D Program of China (No. 2017YFA0305500) (Z.-x.Y.), National Natural Science Foundation of China (No. 61904096) (Z.-x.Y.), Taishan Scholars Program of Shandong Province (No. tsqn201812006) (Z.-x.Y.), Natural Science Foundation of Shandong Province (No. ZR2022JQ05) (Z.-x.Y.) and (No. ZR2022QF025) (F.L.), and a fellowship award from the Research Grants Council of the Hong Kong Special Administrative Region, China (CityU RFS2021–1S04) (J.C.H.).

## Author contributions

Z.-x.Y. conceived the project. Z.-x.Y., F.L., J.C.H., and K.S. designed the experiments and wrote the manuscript. F.L. and Z.S. contributed to the growth of core–shell heterostructure NWs and SEM measurement. D.Q., S.D., L.H., and K.S. performed HRTEM and HAADF STEM. Y.Y. and J.Z. contributed to the XRD measurement. X.Z. and M.W. contributed to the XPS measurement. F.L., Z.S., and J.Z. contributed to the photodetection measurement. X.Z., S.Y., Y.M., Y.T., L.L., and F.C. provided useful discussions. All authors have discussed the results and commented on the manuscript.

## Competing interests

The authors declare no competing interests.
