## [Peer Review File · Nature Communications]

REVIEWER COMMENTS

Reviewer #1 (Remarks to the Author):

This manuscript describes the successful integration of calcogenide (GeS) shell structures on GaSb or GaInAs nanowires. It is very well written and provides a convincing story based on S-passivation of the nanowire side facets followed by the growth of the amorphous GaS material. The material combinations are analyzed by TEM and XPS revealing the material properties.

Although interesting and well-presented, I do not recommend the manuscript for publication in its present form. There are two main points:

1) Optical characterization of detector structures with and without the GeS shell is used to quantify the improvement in performance and to outline the carrier transport mechanisms. The authors present clear improvements in the characteristics. However, the comparison is not very fair. It is well-known that bare nanowires provide not so good performance due to surface effects, including trapping. It is also well established that the surface can be passivated by epitaxial shells with a larger band gap. For a fair comparison, the authors need to compare the GeS shell to an epitaxial shell. Only with this comparison can we evaluate the impact of the GeS material coverage.

2) The IV characteristics for bare GaSb nanowires is compared to data for devices with GeS shells of different thickness. Based on the IV, schematic band diagrams are drawn. This reviewer does not find the argumentation sound and convincing. Most likely the current through the amorphous film is a leakage current (for instance hopping via defect states). It is not unlikely that there is charge trapping at the interface due to states. As such the band structures are misleading. A much better approach would be to form a high quality ohmic contact to the GaSb core and to evaluate the T-dependence of the GeS layers. Only then can we evaluate the band structure.

I strongly recommend the authors to make a revision on the current version with the two points above.

In addition, I wonder why these GaSb nanowires have a triangular cross-section?

Reviewer #2 (Remarks to the Author):

In this manuscript, Liu et al. reports on the synthesis of various core-shell heterostructure NWs based on a lattice-mismatch-free growth strategy of amorphous chalcogenide semiconductors. They demonstrated bidirectional photoresponse and enhanced infrared photodetection properties based on the obtained core-shell heterostructure NWs. This paper addresses some aspects of the lattice mismatch problem at the NW radial interface of core/shell heterostructures. Although the work sounds interesting, the applicability of the lattice-mismatch-free growth strategy of amorphous chalcogenide semiconductors and the performance advantage is rather unclear. Therefore, the manuscript is not acceptable for publication in the current form, it requires major revision.

1. The lattice-mismatch-free growth method is a feasible strategy to solve the lattice mismatch problem of different crystals. However, since chalcogenides are amorphous rather than crystalline shells, is it possible to apply the concept of lattice-mismatch-free growth to chalcogenides?

2. I suggest that the authors demonstrate the application advantages of amorphous materials over crystalline materials to demonstrate the paper's claims for high-quality core/shell heterostructure NWs.

3. In the introduction, the author mentions that the electronic and optoelectronic properties of core-shell heterostructure NWs can be effectively modulated, and introduces related existing work. However, in the abstract and discussion of results, we do not see a comparison of the photodetection performance of the core-shell NWs in the present work with those in the past literature. I suggest that authors should demonstrate the performance and application advantages of the lattice-mismatch-free construction of III-V/chalcogenide core-shell heterostructure nanowires.

4. There is no mention of the stability of the wavelength-dependent bi-directional photoresponse and visible light-assisted infrared photodetection device, and the contrastive performance of the device after some time. It is hoped that the author can provide the research data on device stability.

5. In Figure 4a, why does the difference in shell thickness lead to a difference in positive and negative light response only at 520 nm and 635 nm?

6. The article only explains the phenomenon of why the core-shell heterostructure produces dark current and visible light-assisted near-infrared current but does not explain the influence of shell thickness on this process.

7. For devices with medium shells and thick shells, whether the utilization of the light of 520 nm and 635 nm can also achieve auxiliary enhanced near-infrared photodetection?

8. The authors should provide more detailed experimental information, such as the fine regulation of the core-shell structure.

Reviewer #3 (Remarks to the Author):

Manuscript ID: NCOMMS-23-10752-T

Title: Lattice-mismatch-free construction of III-V/chalcogenide core-shell heterostructure nanowires

Referee report:

The manuscript reports on a general approach for the growth of lattice-mismatch-free core/shell nanowires, obtained combining a crystalline core and an amorphous chalcogenide shell, and their implementation as photodetectors. The results are original, well supported and presented in a clear form. However, it needs some improvements before I can recommend publication in Nature Communication. In particular, the following points should be addressed:

- Concerning the NW growth, in the experimental details is written “the growth substrate with metal catalyst is placed in the downstream zone 3.” I was wondering what is the substrate and which metal catalyst has been used. Please specify (metal type, diameter, dispersion). Also, it would be nice to know if there is any axial growth complementary to the radial growth of the amorphous shell.

- For the GaSb/GeS core/shell nanowires, the EDS data shown in fig. 1 (d-g), 3 (i) and S2 suggest that Ga atoms are present also in the shell. Indeed, while Sb is localized in the central part of the nanowires, with a diameter consistent with the crystalline core diameter, the Ga signal is detected for a larger diameter. Can the authors explain this?

- From Fig. 3 (b, e, h) it seems that the GeS shell is not always homogeneously thick around the nanowire core. Can the authors comment on this, explaining the possible reasons of this shell asymmetry and the possible effects on the performance of the photodetector devices?

Response to Referees' Comments on Manuscript NCOMMS-23-10752-T

We appreciate the referees for considering our manuscript and providing valuable comments. Accordingly, changes have been made in the manuscript, highlighted in red color. Below is our response to the reviewers' comments.

Response to the Reviewers' comments:

Reviewer 1:

Reviewer' comments:

This manuscript describes the successful integration of calcogenide (GeS) shell structures on GaSb or GaInAs nanowires. It is very well written and provides a convincing story based on S-passivation of the nanowire side facets followed by the growth of the amorphous GaS material. The material combinations are analyzed by TEM and XPS revealing the material properties. Although interesting and well-presented, I do not recommend the manuscript for publication in its present form. There are two main points:

Response:

We thank you for the rapid and positive response to our manuscript. We are pleased to submit a revised version of our manuscript based on the valuable suggestions and comments.

(1) Optical characterization of detector structures with and without the GeS shell is used to quantify the improvement in performance and to outline the carrier transport mechanisms. The authors present clear improvements in the characteristics. However, the comparison is not very fair. It is well-know that bare nanowires provide not so good performance due to surface effects, including trapping. It is also well established that the surface can be passivated by epitaxial shells with a larger band gap. For a fair comparison, the authors need to compare the GeS shell to an epitaxial shell. Only with this comparison can we evaluate the impact of the GeS material coverage.

Response:

We appreciate your valuable input. As mentioned by the reviewer, the bare NWs always display not-so-good performance due to the surface charge trappings. Fortunately, the surface passivation technique has been demonstrated as a useful method for manipulating or even reducing surface charge trappings. In this case, the ultra-thin GeS shell and larger bandgap Al₂O₃ shell are grown on the surfaces of GaSb NWs for studying the passivation effect, as suggested by the reviewer. From the high-resolution transmission electron microscope (HRTEM) image of Fig. R11a, with a growth time of 1 s, the shell of as-constructed GaSb/GeS core-shell heterostructure NWs is around 2 nm. As shown in the energy-dispersive X-ray

spectroscopy (EDS) mapping images of Fig. R11b, Ga and Sb dominate the NW core. At the same time, Ge and S dominate the shell. This finding is in line with the result of GaSb/GeS core-shell heterostructure NW with a thicker shell, as shown in Fig. 1. From the broad-spectrum photodetection behavior of Fig. R11c, it is found that GaSb/GeS core-shell heterostructure NW exhibits larger photocurrent compared to that of pristine GaSb NW, which is attributed to the surface passivation effect of ultra-thin GeS shell. At the same time, due to the negative photoresponse caused by the GeS shell, a reduced photocurrent is also observed at the near ultraviolet waveband of 405 nm. In short, the amorphous chalcogenide shells not only overcome the lattice mismatch, but also passivate the surface charge trappings of III-V NWs.

Fig. R11 | Construction of GaSb/GeS core-shell heterostructure NWs with ultra-thin GeS shells and their photodetection behaviors. **a**, HRTEM image of GaSb/GeS core-shell heterostructure NW with ultra-thin shell. **b**, EDS elemental mapping images of Ga, Sb, Al, O. The inset is the corresponding scanning transmission electron microscopy (STEM) image. All the scale bars are 20 nm. **c**, Broad-spectrum photodetection behaviors of pristine GaSb NWs and GaSb/GeS core-shell heterostructure NWs with ultra-thin shell. The laser intensities from 405 nm to 785 nm and 850 nm to 1550 nm are $0.05 \text{ mW}\cdot\text{mm}^{-2}$ and $6.0 \text{ mW}\cdot\text{mm}^{-2}$, respectively.

Beyond the amorphous GeS, the larger bandgap Al_2O_3 is also attempted to passivate the surface charge trappings of GaSb NWs, as shown in Fig. R12. The Al_2O_3 shells are grown on the surfaces of GaSb NWs by the atomic layer deposition method. From HRTEM image and EDS mapping images of Figs. R12a-b, it is found that the GaSb/ Al_2O_3 core-shell heterostructure NW with a shell of 5 nm is successfully constructed. Ga and Sb dominate the NW core. On the other hand, Al and O dominate the shell. From the broad-spectrum photodetection behaviors of Figs. R12c-d, it is found that GaSb NWs with the Al_2O_3 shells of 2 nm and 5 nm exhibit larger photocurrents than the pristine GaSb NW, which is attributed to the surface passivation effect of Al_2O_3 shells. Furthermore, the visible light-assisted infrared photodetection behaviors are also

studied in Figs. R12e-f. When the visible light is on, the infrared photodetection currents can be distinguished hardly. This phenomenon can be attributed to the fact that a large number of carriers generated by visible light act as background carriers for infrared photodetection, which leads to the serious recombination of photogenerated carriers (generated by 1500 nm laser) (*Energy Environ. Sci.* 2018, 11, 417; *Adv. Energy Mater.* 2020, 10, 2000502). The results show that the epitaxial larger bandgap Al_2O_3 shells can passivate the surface charge trappings of GaSb NWs effectively. Compared to the as-constructed III-V/chalcogenide core-shell heterostructure NWs, wavelength-dependent bi-directional photodetection behavior, visible light-assisted infrared photodetection behavior, and faster response times are not observed in GaSb/ Al_2O_3 core-shell heterostructure NWs.

Fig. R12 | Construction of GaSb/ Al_2O_3 core-shell NWs and their photodetection behaviors. a, HRTEM image of GaSb/ Al_2O_3 core-shell NW. **b,** EDS elemental mapping images of Ga, Sb, Al, O. All the scale bars are 20 nm. **c, d,** Broad-spectrum photodetection behaviors of pristine GaSb NWs and GaSb/ Al_2O_3

core-shell NWs with 2 nm and 5 nm Al₂O₃ shell, respectively. The laser intensities from 405 nm to 785 nm and 850 nm to 1550 nm are 0.05 mW·mm⁻² and 6.0 mW·mm⁻², respectively. **e, f**, Infrared photodetection behaviors of GaSb/Al₂O₃ core-shell NWs with 2 nm and 5 nm Al₂O₃ shells under the illuminations of visible light, respectively. The laser intensities of 405 nm and 1550 nm are 0.05 mW·mm⁻² and 5.0 mW·mm⁻², respectively.

In this case, the discussion of “Additionally, as shown in Fig. S11, with an ultra-thin shell of 2 nm (growth time is 1 s), GaSb/GeS core-shell heterostructure NW exhibits larger photocurrents compared to the pristine GaSb NW, which is contributed to the surface passivation effect of ultra-thin GeS shell. At the same time, due to the negative photoresponse caused by the GeS shell, a reduced photocurrent is also observed at the near ultraviolet waveband of 405 nm. This surface passivation effect is also observed in GaSb NWs with the epitaxial shells of larger bandgap Al₂O₃, as shown in Fig. S12. The results show that the epitaxial larger bandgap Al₂O₃ shells can passivate the surface charge trappings of GaSb NWs effectively. Compared to the as-constructed III-V/chalcogenide core-shell heterostructure NWs, wavelength-dependent bi-directional photodetection behavior, visible light-assisted infrared photodetection behavior, and faster response times are not observed in GaSb/Al₂O₃ core-shell heterostructure NW.” has been added to line 24 of page 11 in the revised manuscript. At the same time, Fig. S11 and Fig. S12, along with the corresponding discussions of “The ultra-thin GeS shells are also grown on the surfaces of GaSb NWs for studying the passivation effect. From the HRTEM image of Fig. S11a, with a growth time of 1 s, the shell of as-constructed GaSb/GeS core-shell heterostructure NWs is around 2 nm. As shown in the EDS mapping images of Fig. S11b, Ga and Sb dominate the NW core. At the same time, Ge and S dominate the shell. This result is in line with the result of GaSb/GeS core-shell heterostructure NW with a thicker shell, as shown in Fig. 1. From the broad-spectrum photodetection behavior of Fig. S11c, it is found that GaSb/GeS core-shell heterostructure NW exhibits larger photocurrent compared to that of pristine GaSb NW, which is attributed to the surface passivation effect of ultra-thin GeS shell. At the same time, due to the negative photoresponse caused by the GeS shell, a reduced photocurrent is also observed at the near ultraviolet waveband of 405 nm. In short, the amorphous chalcogenide shells not only overcome the lattice mismatch but also passivate the surface charge trappings of III-V NWs.” and “Beyond the amorphous GeS, the larger bandgap Al₂O₃ is also attempted to passivate the surface charge trappings of GaSb NWs, as shown in Fig. S12. The Al₂O₃ shells grow on the surfaces of GaSb NWs by atomic layer deposition method. The HRTEM image and EDS mapping images of Fig. S12a-b show that GaSb/Al₂O₃ core-shell heterostructure NW with a shell of 5 nm is successfully constructed. Ga and Sb dominate the NW core. On the other hand, Al and O dominate the shell. From the broad-spectrum photodetection behaviors of Fig. S12c-d, it is found that GaSb NWs with the Al₂O₃ shells of 2 nm and 5 nm both exhibit larger photocurrents compared to the pristine GaSb NW, which is attributed to the surface passivation effect of Al₂O₃ shells. Furthermore, the visible light-assisted infrared photodetection behaviors are also studied in Fig. S12e-f. When the visible light is on,

the infrared photodetection currents can be distinguished hardly. This phenomenon can be attributed to the fact that a large number of carriers generated by visible light act as background carriers for infrared photodetection, which leads to the serious recombination of photogenerated carriers (generated by 1500 nm laser)^{7,8}. The results show that the epitaxial larger bandgap Al₂O₃ shells can passivate the surface charge trappings of GaSb NWs effectively. Compared to the as-constructed III-V/chalcogenide core-shell heterostructure NWs, wavelength-dependent bi-directional photodetection behavior, visible light-assisted infrared photodetection behavior, and faster response times are not observed in GaSb/Al₂O₃ core-shell heterostructure NWs.” have been added in the revised supplementary information.

(2) The IV characteristics for bare GaSb nanowires is compared to data for devices with GeS shells of different thickness. Based on the IV, schematic band diagrams are drawn. This reviewer do not find the argumentation sound and convincing. Most likely the current through the amorphous film is a leakage current (for instance hopping via defect states). It is not unlikely that there is charge trapping at the interface due to states. As such the band structures are misleading. A much better approach would be to form a high quality ohmic contact to the GaSb core and to evaluate the T-dependence of the GeS layers. Only then can we evaluate the band structure.

Response:

We thank you for this valuable comment. This work evaluates The band structure by the UPS technique, not by I-V curves. The band diagram in Fig. S6 describes hole carriers' transport process in the as-fabricated GaSb/GeS core-shell heterostructure NW metal-semiconductor-metal (MSM) photodetector. For the pristine GaSb NWs, the I_{DS} of as-fabricated MSM photodetector changes linearly with V_{DS}, demonstrating the typical Ohmic contacts between Ni electrodes and GaSb NWs, as shown in Fig. R13a. With a V_{DS} of 3V, the I_{DS} is around 3.96 μA. This result is in line with the results of previous literature (*Nat. Commun.* 2014, 5, 5249; *Nano Lett.* 2019, 19, 5920). For the GaSb/GeS core-shell heterostructure NWs, the I_{DS} of the as-fabricated MSM photodetector also changes linearly under a small voltage bias and appears saturated I_{DS} under a large voltage bias. With the V_{DS} around 1V, the saturated I_{DS} are around 304, 217, and 12 nA for thin, medium, and thick shell NWs, respectively. Although the I_{DS} are smaller than that of pristine GaSb NWs MSM photodetector, they are unlikely the leakages of amorphous GeS films, which can be deduced from the observation of saturated I_{DS} in the I-V curves. Furthermore, based on the infrared photodetection behaviors of as-constructed GaSb/GeS core-shell heterostructure NWs and InGaAs/GeS core-shell heterostructure NWs, the cores of GaSb and InGaAs act as the main conductive channels. In this case, the transfer path of hole carriers is Ni electrode to GeS shell to GaSb core to GeS shell to Ni electrode, as illustrated in Fig. R13b. The thicker GeS shell gives the larger resistance of the shell, resulting in a smaller I_{DS}. Under a large voltage bias, the mechanism of saturated I_{DS} is presented in Fig. R13c. Taking the positive

bias as an example. Due to the small valence band offset (ΔE_v) between the GeS shell and GaSb core, the hole carriers in the GaSb core will flow across the reverse-biased heterojunction easily under small bias voltage, and the I-V curve presents linear characteristics. However, when the bias voltage increases further, the valence band offset of the reverse bias heterojunction will increase to $(\Delta E_v + e\Delta V)$, which hinders the transfer of hole carriers from the GaSb core to GeS shell, resulting in the saturation of I_{DS} .

Fig. R13 | Electrical properties of GaSb/GeS core-shell heterostructure NWs. **a**, I-V curves of pristine GaSb NW and GaSb/GeS core-shell heterostructure NWs with different shell thicknesses. **b**, Schematic of GaSb/GeS core-shell heterostructure NW MSM photodetector. **c**, Hole carriers transfer process in GaSb/GeS core-shell NWs under a large bias voltage.

Additionally, as suggested by the reviewer, the temperature-dependent electrical properties of the locally etched GaSb/GeS core-shell heterostructure NWs are studied, as shown in Fig. R14. As presented in Fig. R14a of the atomic force microscope (AFM) image, the shell with a thickness of 19.1 nm can be etched by $(CH_3)_4NOH$ aqueous solution (immersed 10 s). In this case, during the device fabrication process, the GeS shell under the metal electrodes can be etched by $(CH_3)_4NOH$ aqueous solution to ensure the expected high-quality Ohmic contacts, as shown in Fig. R14b. Fig. R14c presents the temperature-dependent electrical properties of the locally etched GaSb/GeS core-shell heterostructure NW MSM photodetector. It is found that the I_{DS} changes linearly with V_{DS} , demonstrating the high-quality Ohmic contacts between Ni electrodes and GaSb core, which are comparable to that of pristine GaSb NW MSM photodetector, as shown in Fig. R14d, indicating the GaSb core acts as the main conductive channel. With the temperature decreases, the I_{DS} of both NW MSM photodetectors decrease, which can be attributed to the decrease of carrier concentration in the GaSb core.

Fig. R14 | Temperature-dependent electrical properties of the locally etched GaSb/GeS core-shell heterostructure NW and pristine GaSb NW. **a**, AFM image and height curve of locally etched GaSb/GeS core-shell heterostructure NW. The height curve is extracted from the white dotted lines. **b**, Schematic of locally etched GaSb/GeS core-shell heterostructure NW MSM photodetector. **c**, **d** Temperature-dependent electrical properties of the locally etched GaSb/GeS core-shell heterostructure NW and pristine GaSb NW.

At the same time, we apologize for the misleading caused by the double Y-axis of I-V curves and the improper layouts in Fig. S6. In this case, the I-V curves with a single Y-axis, along with the revised layouts, have been modified in Fig. S7, as shown in Fig. R14. Also, the discussion of “The electrical properties of the as-constructed GaSb/GeS core-shell heterostructure NWs are presented in Fig. S7. For the pristine GaSb NWs, the I_{DS} of as-fabricated MSM photodetector changes linearly with V_{DS} , demonstrating the typical Ohmic contacts between Ni electrodes and GaSb NWs, as shown in Fig. S7a. With a V_{DS} of 3V, the I_{DS} is around 3.96 μA . This result is in line with the results of previous literature^{5,6}. For the GaSb/GeS core-shell heterostructure NWs, the I_{DS} of as-fabricated MSM photodetector also changes linearly under small voltage bias and appears saturated I_{DS} under large voltage bias. With the V_{DS} around 1V, the saturated I_{DS} are around 304, 217, and 12 nA for thin, medium, and thick shell NWs, respectively. Although the I_{DS} are smaller than that of pristine GaSb NWs MSM photodetector, they are unlikely the leakages of amorphous GeS films, which can be deduced from the observation of saturated I_{DS} in the I-V curves. Furthermore, based on the infrared photodetection behaviors of as-constructed GaSb/GeS core-shell heterostructure NWs and InGaAs/GeS core-shell heterostructure NWs, the cores of GaSb and InGaAs act as the main conductive channels. In this case, the transfer path of hole carriers is Ni electrode to GeS shell to GaSb core to GeS shell to Ni electrode, as illustrated in Fig. S7b. The thicker GeS shell gives the larger resistance of the shell, resulting in a smaller I_{DS} . Under large voltage bias, the mechanism of saturated I_{DS} is presented in Fig. S7c. Take the positive bias as an example. Due to the small valence band offset (ΔE_v) between the GeS shell and GaSb core, the hole carriers in the GaSb core will flow across the reverse-biased heterojunction easily under

a small bias voltage, and the I-V curve presents linear characteristics. However, when the bias voltage increases further, the valence band offset of the reverse bias heterojunction will increase to $(\Delta E_v + e\Delta V)$, which hinders the transfer of hole carriers from GaSb core to GeS shell, resulting in the saturation of I_{DS} .” has also been modified in the revised supplementary information.

(3) In addition, I wonder why these GaSb nanowires have a triangular cross-section?

Response:

We appreciate this valuable input. Generally speaking, the sidewalls of zinc blende $\langle 111 \rangle$ -oriented GaSb NW are $\{110\}$ facets, displaying the cross-section of hexagonal shape. In this work, as discussed by the reviewer, the cross-section of as-constructed GaSb/GeS core-shell heterostructure NW is Reuleaux triangular shape. As shown in Fig. R15, the Reuleaux triangular shape has three curved surfaces along $\{112\}A$ (red lines) and three curved surfaces along $\{112\}B$ (green lines). With the slower radial growth rate along $\{112\}B$ direction, the curved surface along $\{112\}A$ dominates the NW sidewall, resulting in the Reuleaux triangular cross-section. As pointed out by Jiang et al., $\{112\}A$ and $\{112\}B$ curved surface facets are decided by the Au-catalyzed vapor-liquid-solid nucleation (*Nano Lett.* 2014, 14, 5865). Indeed, similar triangular cross sections have also been observed in III-V NWs of GaAs NWs, InGaAs NWs, GaAs_{1-x}Sb_x/InP core-shell NWs, InP-InAs core-shell NWs, and InP NWs in the literatures (*Small* 2007, 3, 389; *Nano Lett.* 2013, 13, 5085; *Nano Lett.* 2014, 14, 5865; *Adv. Funct. Mater.* 2015, 25, 5300; *Appl. Phys. Lett.* 2019, 114, 053108; *Nano Energy*, 2020, 71, 104576).

Fig. R15 | Cross-sectional HAADF STEM and FFT images of as-constructed GaSb/GeS core-shell heterostructure NW. a, Cross-sectional HAADF STEM image. **b,** The corresponding FFT pattern. $\{112\}A$ facet, $\langle 422 \rangle$ direction and $\{112\}B$ facet, $\langle 2\bar{2}4 \rangle$ direction are indicated by red and green lines, respectively.

Reviewer 2:

Reviewer' comments:

In this manuscript, Liu et al. reports on the synthesis of various core-shell heterostructure NWs based on a lattice-mismatch-free growth strategy of amorphous chalcogenide semiconductors. They demonstrated bidirectional photoresponse and enhanced infrared photodetection properties based on the obtained core-shell heterostructure NWs. This paper addresses some aspects of the lattice mismatch problem at the NW radial interface of core/shell heterostructures. Although the work sounds interesting, the applicability of the lattice-mismatch-free growth strategy of amorphous chalcogenide semiconductors and the performance advantage is rather unclear. Therefore, the manuscript is not acceptable for publication in the current form, it requires major revision.

Response:

We thank you for the rapid and positive response to our manuscript. We are pleased to submit a revised version of our manuscript based on the valuable suggestions and comments.

(1) The lattice-mismatch-free growth method is a feasible strategy to solve the lattice mismatch problem of different crystals. However, since chalcogenides are amorphous rather than crystalline shells, is it possible to apply the concept of lattice-mismatch-free growth to chalcogenides?

Response:

We appreciate this valuable suggestion. In this work, a versatile strategy is exploited for the lattice-mismatch-free construction of III-V/chalcogenide core-shell heterostructure NWs by simply utilizing the surfactant and amorphous natures of chalcogenide semiconductors. Theoretically, this approach also can be used for lattice-mismatch-free construction of chalcogenide semiconductors core-shell heterostructure NWs. As shown in Fig. R16, the as-expected CdS/GeS core-shell heterostructure NW and the rational band alignment are successfully constructed. From the HRTEM image (Fig. R16a), the as-prepared NW shows apparent contrast between the core and shell. It is also obvious that the shell conformally wraps around the core. The clear lattice fringes are observed with the lattice spacings of 0.65 and 0.36 nm, indicating the good crystallinity of core NW. Noteworthy, no obvious crystal lattice fringes on the surface indicate the shell is amorphous. The elemental compositions of the core and shell are checked by EDS elemental mappings in Fig. R16b. The distribution of element Cd mainly concentrates in the core region, while elements Ge and S are observed in the whole NW, inferring that CdS/GeS core-shell heterostructure NW is successfully constructed. In the end, X-ray diffraction (XRD) further verifies the amorphous shell in Fig. R16c. Furthermore, the as-constructed CdS/GeS core-shell heterostructure NWs exhibit excellent photodetection performance with extended-spectrum detection range and much larger photocurrent compared to that of pristine CdS NWs, as shown in Fig. R16d. The improved photodetection performance

is attributed to the successful construction of type-II p-n core-shell heterostructure, as illustrated in Fig. R16e. Obviously, the developed lattice-mismatch-free construction strategy is efficient and versatile, not only for III-V NWs but also for chalcogenides.

Fig. R16 | Lattice-mismatch-free construction of CdS/GeS core-shell heterostructure NWs and their photodetection behaviors. **a**, HRTEM image of CdS/GeS core-shell heterostructure NW. **b**, EDS elemental mapping images of Cd, Ge, S. All the scale bars are 20 nm. **c**, XRD patterns of CdS NWs and as-constructed core-shell heterostructure NWs. **d**, Photodetection behaviors of pristine CdS NW and CdS/GeS core-shell heterostructure NW. **e**, Schematic for type-II CdS/GeS core-shell NWs photodetector.

In this case, the discussion of “Furthermore, as shown in Fig. S17, the lattice-mismatch-free construction approach can also be used to grow chalcogenides core-shell heterostructure NWs. Owing to the rational band alignment, the as-constructed CdS/GeS core-shell heterostructure NWs exhibit excellent photodetection performance with extended-spectrum detection range and much larger photocurrent compared to that of pristine CdS NWs.” has been added to line 24 of page 15 in the revised manuscript. At the same time, Fig. S17, along with the corresponding discussion of “In this work, a versatile strategy is exploited for the lattice-mismatch-free construction of III-V/chalcogenide core-shell heterostructure NWs by simply utilizing the surfactant and amorphous natures of chalcogenide semiconductors. Theoretically, this approach also can be used for lattice-mismatch-free construction of chalcogenide semiconductors core-shell heterostructure NWs. As shown in Fig. S17, the as-expected CdS/GeS core-shell heterostructure NW and the rational band alignment are successfully constructed. From the HRTEM image (Fig. S17a), the as-prepared NW shows apparent contrast between the core and shell. It is also obvious that the shell conformally wraps around the core. The clear lattice fringes are observed with the lattice spacings of 0.65

and 0.36 nm, indicating the good crystallinity of core NW. Noteworthy, no obvious crystal lattice fringes on the surface indicate the shell is amorphous. The elemental compositions of the core and shell are checked by EDS elemental mappings in Fig. S17b. The distribution of element Cd mainly concentrates in the core region, while elements Ge and S are observed in the whole NW, inferring that CdS/GeS core-shell heterostructure NW is successfully constructed. In the end, X-ray diffraction (XRD) further verifies the amorphous shell in Fig. S17c. Furthermore, the as-constructed CdS/GeS core-shell heterostructure NWs exhibit excellent photodetection performance with extended-spectrum detection range and much larger photocurrent compared to that of pristine CdS NWs, as shown in Fig. S17d. The improved photodetection performance is attributed to the successful construction of type-II p-n core-shell heterostructure, as illustrated in Fig. S17e. Obviously, the developed lattice-mismatch-free construction strategy is efficient and versatile, not only for III-V NWs but also for chalcogenides.” have been added in the revised supplementary information.

(2) I suggest that the authors demonstrate the application advantages of amorphous materials over crystalline materials to demonstrate the paper’s claims for high-quality core/shell heterostructure NWs.

Response:

We thank you for the important comment. Lattice mismatch is challenging the construction and utilization of III-V core-shell heterostructure NWs. Although growing a very thin shell layer or selecting the constituent materials with very close lattice constants can minimize the adverse impact of lattice mismatch, it is still not easy to construct III-V core-shell heterostructure NWs even by using MBE or MOCVD technologies. In this work, by simply utilizing the surfactant and amorphous natures of chalcogenide semiconductors, III-V/chalcogenide core-shell heterostructure NWs are constructed successfully by the simple CVD technology. As a contrast, the crystalline III-V semiconductor of GaAs is adopted to construct III-V core-shell heterostructure NWs, as shown in Fig. R17. From the scanning electron microscope (SEM) and TEM images of Figs. R17 a&b, the as-constructed NWs exhibit serious surface coatings. The poor crystallinity of as-constructed NWs is further revealed by HRTEM (Fig. R17c). The compositions of as-constructed NW are then evaluated by EDS mappings, as presented in Fig. R17d. It is found that all the elements of Ga, Sb, and As are distributed in the whole as-constructed NW. This result may be caused by the same source temperatures of GaAs and GaSb (both heated at 750°) during the growth process of GaAs shells. From Fig. R17e, the as-fabricated GaSb/GaAs core-shell NW photodetector exhibits unsatisfactory photodetection performance with weak and noisy photocurrents, which may result from the heavy surface coatings. Obviously, owing to the lattice mismatch between crystalline GaSb and GaAs, the as-constructed core-shell heterostructure NWs show serious surface coating and unsatisfactory photodetection performance.

Fig. R17 | Construction of GaSb/GaAs core-shell heterostructure NWs and their photodetection behaviors. **a-c**, The SEM, TEM, HRTEM images of as-constructed GaSb/GaAs core-shell heterostructure NWs. The insets of (c) show the corresponding FFT patterns in the shell and core regions, respectively. **d**, STEM image and EDS elemental mappings of Ga, Sb, As. All the scale bars are 100 nm. **e**, Broad-spectrum photodetection behavior of GaSb/GaAs core-shell heterostructure NW. The laser intensities from 405 nm to 785 nm and 850 nm to 1550 nm are $0.05 \text{ mW}\cdot\text{mm}^{-2}$ and $6.0 \text{ mW}\cdot\text{mm}^{-2}$, respectively.

(3) In the introduction, the author mentions that the electronic and optoelectronic properties of core-shell heterostructure NWs can be effectively modulated, and introduces related existing work. However, in the abstract and discussion of results, we do not see a comparison of the photodetection performance of the core-shell NWs in the present work with those in the past literature. I suggest that authors should demonstrate the performance and application advantages of the lattice-mismatch-free construction of III-V/chalcogenide core-shell heterostructure nanowires.

Response:

We thank you for the important input. Many efforts have been focused on constructing III-V core-shell heterostructure NWs, but the study on their photodetection behaviors is still rare, as shown in Table R3. In contrast, by adopting the developed lattice-mismatch-free construction approach, a variety of III-V/chalcogenide core-shell heterostructure NWs with desired band alignment can be constructed, along with the intriguing photodetection behaviors, such as wavelength-dependent bi-directional photoresponse and visible light-assisted infrared photodetection behaviors. At the same time, the R of the lattice-mismatch-

free constructed III-V/chalcogenide core-shell heterostructure NWs photodetectors in this work are much better than that of the reported photodetectors. In a word, the lattice-mismatch-free construction of core-shell heterostructure NWs with rational band alignment is an effective strategy to realize next-generation high-performance optoelectronics devices.

Table R3. Photodetection performances comparison of III-V core-shell heterostructure NWs.

Core-shell NWs	Growth method	Heterostructure type	Photoresponse behavior	R (A/W)	D^* (Jones)	Response time (t_r/t_a)	Ref.
GaAs/AlGaAs	MOCVD	Type-I	Positive	0.57 (@855 nm)	7.2×10^{10}		[1]
GaAs _{1-x} Sb _x /InAs	MBE	Type-II p-n	Positive	0.12 (@1310 nm)	-	0.45 ms/0.86 ms (633nm)	[2]
InAs/AlSb	MBE	Type-II p-n	Negative	-	-	-	[3]
GaAs/AlGaAs	MOVPE	Type-I	Positive	10^{-4} (@800 nm)	-	5 ps	[4]
In-Rich InGaAs	MBE	-	Positive	5.75 (@1550 nm)	-	-	[5]
GaSb/GeS	CVD	Type-I p-p	Wavelength dependent bi-directional	400 (@1550 nm)	9.8×10^{10}	12 ms/8 ms (1550 nm)	This work
InGaAs/GeS	CVD	Type-II p-n	Positive	61.2 (@850 nm)	6.8×10^{11}	3.3 ms/2.9 ms (850 nm)	This work

In this case, the discussion of “Up to now, many efforts have been focused on constructing III-V core-shell heterostructure NWs. However, the study on their photodetection behaviors is still rare, as shown in Table S1⁵⁰⁻⁵⁴. On the contrary, by adopting the developed lattice-mismatch-free construction approach, a variety of III-V/chalcogenide core-shell heterostructure NWs with desired band alignment can be constructed, along with the intriguing photodetection behaviors.” has been added on line 19 of page 15 in the revised manuscript. At the same time, Table S1 has been added to the revised supplementary information.

(4) *There is no mention of the stability of the wavelength-dependent bi-directional photoresponse and visible light-assisted infrared photodetection device, and the contrastive performance of the device after some time. It is hoped that the author can provide the research data on device stability.*

Response:

We appreciate the valuable suggestion. As discussed by the reviewer, the stability of the as-fabricated GaSb/GeS core-shell heterostructure NWs photodetector is important for further practical application. As presented in Fig. R18a, the as-fabricated photodetector still exhibits stable wavelength-dependent bi-directional photodetection behavior, which displays a negative photoresponse in the wavelength of 405-785 nm and a positive photoresponse in the wavelength of 850-1550 nm after being stored in an atmospheric environment for 10 days and 30 days. The photocurrent attenuations are less than 10% and 20% for 10 days and 30 days, respectively. As shown in Fig. R18b, the as-fabricated photodetector also shows the visible

light-assisted infrared photodetection behavior after being stored in an atmospheric environment for 30 days. It is found that when the assisted light of 405 nm is on, the dark current of the infrared photodetector is significantly suppressed, and the photocurrent increases obviously. In short, benefiting the as-constructed core-shell nanostructure, the as-fabricated photodetector exhibits stable, repeatable, and robust photodetection performance, which promises the application in further optoelectronic devices.

Fig. R18 | Stability of as-fabricated GaSb/GeS core-shell heterostructure NW photodetector. **a**, Broad-spectrum photodetection behavior of GaSb/GeS core-shell heterostructure NW after being stored in an atmospheric environment for 10 days and 30 days. The laser intensities from 405 nm to 785 nm and from 850 nm to 1550 nm are 0.05 mW·mm⁻² and 6.0 mW·mm⁻², respectively. **b**, Visible light-assisted infrared photodetection performance of GaSb/GeS core-shell heterostructure NW after being stored in an atmospheric environment for 30 days.

In this regard, the discussion of “Furthermore, benefiting from the as-constructed core-shell nanostructure, the as-fabricated photodetector exhibits stable wavelength-dependent bi-directional photodetection and visible light-assisted infrared photodetection behaviors after being stored in the atmospheric environment for 30 days, as shown in Fig. S13.” has been added to line 7 of page 12 in the revised manuscript. At the same time, Fig. S13, along with the corresponding discussion of “As presented in Fig. S13a, the as-fabricated photodetector still exhibits stable wavelength-dependent bi-directional photodetection behavior, which displays a negative photoresponse in the wavelength of 405-785 nm and a positive photoresponse in the wavelength of 850-1550 nm after being stored in an atmospheric environment for 10 days and 30 days. The photocurrent attenuations are less than 10% and 20% for 10 days and 30 days, respectively. As shown in Fig. S13b, the as-fabricated photodetector also shows the visible light-assisted infrared photodetection

behavior after being stored in an atmospheric environment for 30 days. It is found that when the assisted light of 405 nm is on, the dark current of the infrared photodetector is significantly suppressed, and the photocurrent increases obviously. In short, benefitting to the as-constructed core-shell nanostructure, the as-fabricated photodetector exhibits stable, repeatable, and robust photodetection performance, which promises the application in further optoelectronic devices.” have been added in the revised supplementary information.

(5) In Figure 4a, why does the difference in shell thickness lead to a difference in positive and negative light response only at 520 nm and 635 nm?

Response:

We thank the reviewer for this careful input. After a careful check of Fig. 4a, it is found that the shell thickness plays an important role in the bi-directional photodetection behavior in the 405-785 nm wavelength. In the wavelength of 405-785 nm, the positive photoresponse switches to negative photoresponse, with the shell thickness increasing from 11.3 ± 2.0 nm (thin shell) to 14.7 ± 1.9 nm (medium shell) and 19.5 ± 4.7 nm (thick shell). At the same time, only positive photoresponse is observed in the wavelength of 850-1550 nm. The interesting bi-directional photodetection behavior observed with different shell thicknesses is attributed to the competition between the GeS shell's negative photoresponse and the GaSb core's positive photoresponse. Under visible light illumination (< 785 nm), both GeS shell and GaSb core can absorb light. The absorption of visible light by the GeS shell results in a negative photoresponse, while the absorption of visible light by the GaSb core results in a positive photoresponse. Obviously, with a thin shell, the proportion of visible light absorbed by GeS is limited, and the visible light is mainly absorbed by GaSb, resulting in a positive photoresponse. With a thick shell, the proportion of visible light absorbed by GeS increases, while that absorbed by GaSb decreases, resulting in a negative photoresponse.

Hence, the discussion of “Obviously, the negative and positive photoresponse coexist and compete in the as-constructed core-shell heterostructure NW under the illumination of visible light, resulting in the interesting shell thickness dependent bi-directional photodetection behavior. With a thin shell, the proportion of visible light absorbed by GeS is limited, and the visible light is mainly absorbed by GaSb, resulting in a positive photoresponse. With a thick shell, the proportion of visible light absorbed by GeS increases, while that absorbed by GaSb decreases, resulting in the negative photoresponse.” has been added on line 24 of page 12 in the revised manuscript.

(6) The article only explains the phenomenon of why the core-shell heterostructure produces dark current and visible light-assisted near-infrared current but does not explain the influence of shell thickness on this process.

Response:

We appreciate this valuable input. Indeed, the shell thickness also plays an important role in visible light-assisted behavior. As shown in Fig. R19a, when the visible light (405 nm) is on, the infrared photodetection current is suppressed in the thin shell core-shell NWs photodetector. This suppression is because the absorption of visible light is dominated by the GaSb core in the thin shell core-shell NWs. This way, the photogenerated carriers generated by 1500 nm laser are annihilated in the large number of carriers generated by visible light, resulting in suppressed photocurrents. With a medium shell, the visible light is mainly absorbed by the GeS shell. When the visible light is on, the dark current of the photodetector for infrared light is reduced due to the photogenerated electrons in the GeS shell recombining with the free holes in the GaSb core, as shown in Fig. 4e. Meanwhile, the trapped photogenerated holes in the GeS shell lead to the field-effect passivation (*Appl. Phys. Lett.* 2014, 104, 232903; *Nano Res.* 2015, 8, 673), which would inhibit the recombination of the infrared light excited electron-hole pairs in the GaSb core, ultimately leading to the improvement of infrared photodetection current. With a thick shell, the dark current is also decreased with visible light, as shown in Fig. R19b. But barely any improvement of infrared photodetection current is observed, possibly due to the thick shell weakening the field-effect passivation effect.

Fig. R19. The visible light-assisted infrared photodetection performance of NWs with (a) a thin shell and (b) a thick shell.

In this case, the discussion of “Meanwhile, the trapped photogenerated holes in the GeS shell leads to the photogating effect, which would passivate the interface defects (field-effect passivation) and prolong the lifetime of infrared light excited electron-hole pairs in the GaSb core, ultimately leading to the improvement of infrared photodetection performances⁴⁵⁻⁴⁷.” has been revised on line 7 of page 13 in the revised manuscript. The discussion of “The shell thickness also plays an important role in visible light-assisted behavior. As shown in Fig. S14a, when the visible light (405 nm) is on, the infrared photodetection currents are suppressed in the thin shell core-shell NWs photodetector. This suppression is because the absorption of visible light is dominated by the GaSb core in the thin shell core-shell NWs. In this case, the photogenerated carriers generated by 1500 nm laser are annihilated in the large number of carriers generated

by visible light, resulting in suppressed photocurrents. With a medium shell, the visible light is mainly absorbed by the GeS shell. When the visible light is on, the dark current of the photodetector for infrared light is reduced due to the photogenerated electrons in the GeS shell recombining with the free holes in the GaSb core, as shown in Fig. 4e. Meanwhile, the trapped photogenerated holes in the GeS shell lead to the field-effect passivation^{9,10}, which would inhibit the recombination of the infrared light excited electron-hole pairs in the GaSb core, ultimately leading to the improvement of infrared photodetection currents. With a thick shell, the dark current is also decreased with visible light, as shown in Fig. S14b. But barely any improvement of infrared photodetection current is observed, possibly due to the thick shell weakening the field-effect passivation effect.” have been added in the revised supplementary information.

(7) For devices with medium shells and thick shells, whether the utilization of the light of 520 nm and 635 nm can also achieve auxiliary enhanced near-infrared photodetection?

Response:

We thank you for this careful comment. Based on the mechanism of visible light-assisted behavior, both 520 nm and 635 nm lights also can improve the infrared photodetection performance of photodetector fabricated by GaSb/GeS core-shell heterostructure NWs with an appropriate shell thickness. As shown in Fig. R20a, with a medium shell, when assisted light of 520 nm is on, the dark current of the infrared photodetector decreases from 269 nA to 187 nA. At the same time, the photocurrent increases from 10 nA to 27 nA. When auxiliary light of 635 nm is on (Fig. R20b), the dark current of the infrared photodetector decreases from 266 nA to 216 nA, and the photocurrent increases from 11 nA to 22 nA. For the photodetector with a thick shell, with the assistance of visible lights of 520 nm and 635 nm, the dark currents of the photodetectors for infrared light are reduced, as shown in Fig. R20c and R20d. But barely any improvement of infrared photodetection currents is observed, similar to 405 nm assisted light (Fig. R19b).

Fig. R20 | Visible light-assisted infrared photodetection performance of GaSb/GeS core-shell heterostructure NWs. **a-b**, The visible light-assisted infrared photodetection performance of NW with a medium shell, where 520 nm and 635 nm light act as assisted light, respectively. **c-d**, The visible light-assisted infrared photodetection performance of NW with a thick shell, where 520 nm and 635 nm light act as assisted light, respectively.

In this regard, the discussion of “This visible light-assisted behavior is also observed under the illuminations of 520 nm and 635 nm lasers, as shown in Fig. S9.” has been added to line 19 of page 11 in the revised manuscript. At the same time, Fig. S9, along with the corresponding discussion of “Based on the mechanism of visible light-assisted behavior, both 520 nm and 635 nm lights also can improve the infrared photodetection performance of photodetector fabricated by GaSb/GeS core-shell heterostructure NWs with an appropriate shell thickness. As shown in Fig. S9a, with a medium shell, when assisted light of 520 nm is on, the dark current of the infrared photodetector decreases from 269 nA to 187 nA. At the same time, the photocurrent increases from 10 nA to 27 nA. When auxiliary light of 635 nm is on (Fig. S9b), the dark current of the infrared photodetector decreases from 266 nA to 216 nA, and the photocurrent increases from 11 nA to 22 nA. For the photodetector with a thick shell, with the assistance of visible lights of 520 nm and 635 nm, the dark currents of the photodetectors for infrared light are reduced, as shown in Fig. S9c and S9d. But barely any improvement of infrared photodetection currents is observed, similar to that of 405 nm assisted light (Fig. S14b).” have been added in the revised supplementary information.

(8) *The authors should provide more detailed experimental information, such as the fine regulation of the core-shell structure.*

Response:

We appreciate the valuable comment. A table about the growth details of core-shell heterostructure NWs, including the mass of sources, the growth substrates, the thickness of metal catalyst, the position of source and growth substrate, the flow of carrier gas, the pressure of CVD system and the growth temperatures and times have been added to the revised manuscript.

Table R4. Growth details of core-shell heterostructure NWs.

Core-shell NWs	Catalyst (nm)	Growth of core NWs					Growth of shells			
		Source (g)	Sulfur (g)	S.T. (°C)	G.T. (°C)	Carrier gas (sccm)	Source (g)	S.T. (°C)	G.T. (°C)	Carrier gas (sccm)
GaSb/GeS	Au/1.0	GaSb/0.4	0.4	750	560	H ₂ /200	GeS/0.1	500	320	H ₂ /200
GaAs/GeS	Au/1.0	GaAs/0.4	-	800	600	H ₂ /200	GeS/0.1	500	320	H ₂ /200
InGaAs/GeS	Ni/1.0	(InAs:GaAs =1:1)/0.4	-	810	610	H ₂ /200	GeS/0.1	500	320	H ₂ /200
GaSb/GeSe	Au/1.0	GaSb/0.4	0.4	750	560	H ₂ /200	GeSe/0.1	550	320	H ₂ /200
CdS/GeS	Au/1.0	CdS/0.1	-	750	540	Ar/50	GeS/0.1	500	320	H ₂ /200
GaSb/GaAs	Au/1.0	GaSb/0.4	0.4	750	560	H ₂ /200	GaAs/0.4	750	560	H ₂ /200

Note: S.T.: Source temperature; G.T.: Growth temperature; Growth substrate: Si/SiO₂; Pressure of CVD system: 6×10⁻³ Torr; Position of source and growth substrate: source powders of shell semiconductors (such as GeS, GeSe, and GaAs) are placed in upstream zone 1, source powders of core semiconductors (such as GaSb, GaAs, InGaAs, and CdS) are placed in midstream zone 2, the growth substrate is placed in the downstream zone 3, and sulfur powders are placed between zone 2 and 3.

Therefore, the discussion of “**The growth details of core-shell heterostructure NWs can be found in Table S2.**” has been added to line 27 of page 16 in the revised manuscript.

Reviewer 3:

Reviewer' comments:

The manuscript reports on a general approach for the growth of lattice-mismatch-free core/shell nanowires, obtained combining a crystalline core and an amorphous chalcogenide shell, and their implementation as photodetectors. The results are original, well supported and presented in a clear form. However, it needs some improvements before I can recommend publication in Nature Communication. In particular, the following points should be addressed:

Response:

We thank you for the positive response to our manuscript. We are pleased to submit a revised version of our manuscript based on the valuable suggestions and comments.

(1) Concerning the NW growth, in the experimental details is written “the growth substrate with metal catalyst is placed in the downstream zone 3.” I was wondering what is the substrate and which metal

catalyst has been used. Please specify (metal type, diameter, dispersion). Also, it would be nice to know if there is any axial growth complementary to the radial growth of the amorphous shell.

Response:

We appreciate the useful suggestions. In this work, the Si/SiO₂ (50 nm thick thermally grown) is used as the growth substrate. The thermal evaporation of 1 nm Au film is employed as the growth catalyst. This catalyst's fabrication is similar to the experimental conditions in our previous work (*Nat. Commun.* 2014, 5, 5249). With a thickness of 1 nm, the Au catalyst particles are uniformly distributed on the Si/SiO₂ substrate with an average diameter of 17.8 ± 4.9 nm. Furthermore, a table about the growth details of core-shell heterostructure NWs, including the mass of sources, the growth substrates, the thickness of metal catalyst, the position of source and growth substrate, the flow of carrier gas, the pressure of CVD system and the growth temperatures and times have been added to the revised manuscript.

Table R5. Growth details of core-shell heterostructure NWs.

Core-shell NWs	Catalyst (nm)	Growth of core NWs					Growth of shells			
		Source (g)	Sulfur (g)	S.T. (°C)	G.T. (°C)	Carrier gas (sccm)	Source (g)	S.T. (°C)	G.T. (°C)	Carrier gas (sccm)
GaSb/GeS	Au/1.0	GaSb/0.4	0.4	750	560	H ₂ /200	GeS/0.1	500	320	H ₂ /200
GaAs/GeS	Au/1.0	GaAs/0.4	-	800	600	H ₂ /200	GeS/0.1	500	320	H ₂ /200
InGaAs/GeS	Ni/1.0	(InAs:GaAs=1:1)/0.4	-	810	610	H ₂ /200	GeS/0.1	500	320	H ₂ /200
GaSb/GeSe	Au/1.0	GaSb/0.4	0.4	750	560	H ₂ /200	GeSe/0.1	550	320	H ₂ /200
CdS/GeS	Au/1.0	CdS/0.1	-	750	540	Ar/50	GeS/0.1	500	320	H ₂ /200
GaSb/GaAs	Au/1.0	GaSb/0.4	0.4	750	560	H ₂ /200	GaAs/0.4	750	560	H ₂ /200

Note: S.T.: Source temperature; G.T.: Growth temperature; Growth substrate: Si/SiO₂; Pressure of CVD system: 6×10⁻³ Torr; Position of source and growth substrate: source powders of shell semiconductors (such as GeS, GeSe, and GaAs) are placed in upstream zone 1, source powders of core semiconductors (such as GaSb, GaAs, InGaAs, and CdS) are placed in midstream zone 2, the growth substrate is placed in the downstream zone 3, and sulfur powders are placed between zone 2 and 3.

For checking the possibility of axial growth, the tails of as-constructed GaSb/GeS core-shell heterostructure NWs are observed by TEM. As shown in Fig. R21, all the NWs exhibit conformal core-shell heterostructure. No axial growth of amorphous GeS is observed. This finding is attributed to the intermediary role of chalcogens, which promotes the conformal growth of amorphous chalcogenide around III-V NWs. At the same time, the GeS shells are homogeneous around GaSb NW cores.

Fig. R21 | TEM images of the as-constructed GaSb/GeS core-shell heterostructure NWs. All the scale bars are 50 nm.

In this case, the discussion of “**The growth details of core-shell heterostructure NWs can be found in Table S2.**” has been added to line 27 of page 16 in the revised manuscript. The discussion of “**For checking the possibility of axial growth, the tails of as-constructed GaSb/GeS core-shell heterostructure NWs are observed by TEM. As shown in Fig. S3, all the NWs exhibit conformal core-shell heterostructure. No axial growth of amorphous GeS is observed.**” has been added to line 22 of page 5 in the revised manuscript. At the same time, **Fig. S3** and **Table S2** have been added to the revised supplementary information.

(2) For the GaSb/GeS core/shell nanowires, the EDS data shown in fig. 1 (d-g), 3 (i) and S2 suggest that Ga atoms are present also in the shell. Indeed, while Sb is localized in the central part of the nanowires, with a diameter consistent with the crystalline core diameter, the Ga signal is detected for a larger diameter. Can the authors explain this?

Response:

We thank you for this careful comment. Because the cross-sectional TEM specimens are prepared by using a focused Ga⁺ ion beam system, Ga would exist in the entire NWs, as shown in the EDS mappings of Fig.1 and S2. In fact, owing to the sulfur surfactant being used in the NWs growth process, the unsaturated surface-terminated Sb is effectively stabilized to form the stable S-Sb bond, which results in a little larger distribution diameter of Ga than Sb. This result is in line with the previous result (*Nat. Commun.* 2014, 5, 5249) and can also be observed in the EDS mapping images of pristine GaSb NWs in Fig. R22. Without

the sulfur surfactant, the diameter of Ga is similar to that of As, as shown in GaAs/GeS (Fig. 3c) and InGaAs/GeS (Fig. 3f) core-shell heterostructure NWs.

Fig. R22 | EDS elemental mapping images of pristine GaSb NWs. a, STEM image of pristine GaSb NWs. **b-d,** EDS elemental mappings of S, Ga, and Sb. All the scale bars are 20 nm.

In this case, the discussion of “Owing to the Ga⁺ ion is used in the preparation of cross-sectional TEM specimens, Ga exists in the entire NW.” has been added to line 29 of page 5 in the revised manuscript. The discussion of “In fact, owing to the sulfur surfactant being used in the NWs growth process, the unsaturated surface-terminated Sb is effectively stabilized to form the stable S-Sb bond, which results in a little larger distribution diameter of Ga than Sb^{27,34,38}, as shown in Fig. 3i.” has been added to line 10 of page 10 in the revised manuscript.

(3) From Fig. 3 (b, e, h) it seems that the GeS shell is not always homogeneously thick around the nanowire core. Can the authors comment on this, explaining the possible reasons of this shell asymmetry and the possible effects on the performance of the photodetector devices?

Response:

We appreciate the reviewer for this careful and interesting comment. After a careful check, the shells look inhomogeneous around the NW cores in the HRTEM images (Figs. 3b, 3e & 3h). However, in the cross-sectional HAADF STEM images of Fig. 1 and Fig. S2 and low-resolution STEM images of Figs. 3c, 3f & 3i, the shells are homogeneous around the NW cores. In this case, the inhomogeneous may be caused by the shooting angle during the observation of HRTEM. Under the smaller view, a slight sample incline will

cause inhomogeneity observation. This conclusion is also verified by the low-resolution TEM images of other 10 GaSb/GeS core-shell heterostructure NWs, as shown in Fig. R21. Additionally, the photodetection performance of several InGaAs/GeS core-shell heterostructure NWs photodetectors are studied in Fig. R23. Obviously, all the photodetectors exhibit excellent broad-spectrum photodetection performance, indicating the uniformity of as-constructed core-shell NWs.

Fig. R23 | Photodetection behaviors of other 3 InGaAs/GeS core-shell heterostructure NWs.

In this case, the discussion of “The shells look inhomogeneous around the NW cores in the HRTEM images of Figs. 3b, 3e & 3h. However, in the cross-sectional HAADF STEM images of Fig. 1 and Fig. S2, low-resolution STEM images of Figs. 3c, 3f & 3i and TEM images of Figs. S3 & S4, the shells are homogeneous around the NW cores. The inhomogeneity may be caused by the shooting angle during the observation of HRTEM. Under the smaller view, a slight incline of the sample will cause the observation of inhomogeneity.” has been added to line 3 of page 9 in the revised manuscript.

Changes List in the Revised Manuscript

1. On line 2 of page 2, “Growing high-quality core-shell heterostructure nanowires (NWs) is still challenging due to the lattice mismatch issue at the radial interface.” has been revised.
2. On line 22 of page 5, “For checking the possibility of axial growth, the tails of as-constructed GaSb/GeS core-shell heterostructure NWs are observed by TEM. As shown in Fig. S3, all the NWs exhibit conformal core-shell heterostructure. No axial growth of amorphous GeS is observed.” has been added.
3. On line 29 of page 5, “Owing to the Ga⁺ ion is used in the preparation of cross-sectional TEM specimens, Ga looks like in the whole NW.” has been added.
4. On line 18 of page 8, “As shown in the scanning electron microscope (SEM) images in Figs. 3a, 3d & 3g,” has been revised.
5. In page 9, “Importantly, the core-shell integration of the above semiconductors can realize a variety of heterostructures with different carrier transport characteristics, as shown in Fig. S4. They include the type-I heterostructures (e.g., GaSb/GeS and GaSb/GeSe) and the type-II heterostructures (e.g., GaAs/GeS and InGaAs/GeS), which are promising for building various high-performance optoelectronics devices¹³.” has been deleted.
6. On line 1 of page 9, “Based on the high-resolution TEM (HRTEM) images (Figs. 3b, 3e & 3h), all NWs are core-shell nanostructures, along with the clear contrasts between the cores and shells. The shells look inhomogeneous around the NW cores in the HRTEM images of Figs. 3b, 3e & 3h. However, in the cross-sectional HAADF STEM images of Fig. 1 and Fig. S2, low-resolution STEM images of Figs. 3c, 3f & 3i and TEM images of Figs. S3 & S4, the shells are homogeneous around the NW cores. The inhomogeneity may be caused by the shooting angle during the observation of HRTEM. Under the smaller view, a slight incline of the sample will cause the observation of inhomogeneity.” has been added.
7. On line 13 of page 9, “Importantly, the core-shell integration of the above semiconductors can realize a variety of heterostructures, as shown in Fig. S5. They include the type-I heterostructures (e.g., GaSb/GeS) and the type-II heterostructures (e.g., GaAs/GeS, InGaAs/GeS, and GaSb/GeSe), which are promising for building various high-performance optoelectronics devices¹³.” has been added.
8. On line 5 of page 10, “The compositions are then evaluated by EDS elemental mappings as presented in Figs. 3c, 3f & 3i.” has been revised.
9. On line 10 of page 10, “In fact, owing to the surfactant of sulfur is used in the NWs growth process, the unsaturated surface-terminated Sb is effectively stabilized to form the stable S-Sb bond, which results in the little larger distribution diameter of Ga than Sb^{27,34,38}, as shown in Fig 3i.” has been added.
10. On line 19 of page 11, “This visible light-assisted behavior is also observed under the illuminations of 520 nm and 635 nm lasers, as shown in Fig. S9.” has been added.

11. On line 24 of page 11, “Additionally, as shown in Fig. S11, with an ultra-thin shell of 2 nm (growth time is 1 s), GaSb/GeS core-shell heterostructure NW exhibits larger photocurrents compared to the pristine GaSb NW, which is contributed to the surface passivation effect of ultra-thin GeS shell. At the same time, due to the negative photoresponse caused by the GeS shell, a reduced photocurrent is also observed at the near ultraviolet waveband of 405 nm. This surface passivation effect is also observed in GaSb NWs with the epitaxial shells of larger bandgap Al₂O₃, as shown in Fig. S12. The results show that the epitaxial larger bandgap Al₂O₃ shells can passivate the surface charge trappings of GaSb NWs effectively. Compared to the as-constructed III-V/chalcogenide core-shell heterostructure NWs, wavelength-dependent bi-directional photodetection behavior, visible light-assisted infrared photodetection behavior, and faster response times are not observed in GaSb/Al₂O₃ core-shell heterostructure NW. Furthermore, benefiting from the as-constructed core-shell nanostructure, the as-fabricated photodetector exhibits stable wavelength-dependent bi-directional photodetection and visible light-assisted infrared photodetection behaviors after being stored in the atmospheric environment for 30 days, as shown in Fig. S13.” has been added.
12. In page 12, “The band structures of the amorphous GeS shell and crystalline GaSb core are verified by ultraviolet photoelectron spectra (UPS), as shown in Fig. S9.” has been deleted.
13. On line 13 of page 12, “With the bandgaps of 0.70 eV and 1.50 eV for GaSb core and GeS shell, both of core and shell can absorb the visible light (< 827 nm) and generate electron-hole pairs (processes I and I’).” has been revised.
14. On line 26 of page 12, “Obviously, the negative and positive photoresponse coexist and compete in the as-constructed core-shell heterostructure NW under the illumination of visible light, resulting in the interesting shell thickness dependent bi-directional photodetection behavior. With a thin shell, the proportion of visible light absorbed by GeS is limited, and the visible light is mainly absorbed by GaSb, resulting in a positive photoresponse. With a thick shell, the proportion of visible light absorbed by GeS increases, while that absorbed by GaSb decreases, resulting in a negative photoresponse.” has been added.
15. On line 7 of page 13, “Meanwhile, the trapped photogenerated holes in the GeS shell leads to the photogating effect, which would passivate the interface defects (field-effect passivation) and prolong the lifetime of infrared light excited electron-hole pairs in the GaSb core, ultimately leading to the improvement of infrared photodetection performances⁴⁵⁻⁴⁷. It is worth pointing out that the shell thickness also plays an important role in the visible light-assisted behavior, as shown in Fig. S14. In a word, a careful design of appropriate shell thickness is necessary for high-performance bi-directional photodetection and visible light-assisted photodetection.” has been added.

-
16. On line 19 of page 15, “Up to now, many efforts have been focused on constructing III-V core-shell heterostructure NWs, but the study on their photodetection behaviors is still rare, as shown in Table S1⁵⁰⁻⁵⁴. On the contrary, by adopting the developed lattice-mismatch-free construction approach, various III-V/chalcogenide core-shell heterostructure NWs with desired band alignment can be constructed, along with intriguing photodetection behaviors. Furthermore, as shown in Fig. S17, the lattice-mismatch-free construction approach can also be used to grow chalcogenides core-shell heterostructure NWs. Owing to the rational band alignment, the as-constructed CdS/GeS core-shell heterostructure NWs exhibit excellent photodetection performance with extended-spectrum detection range and much larger photocurrent compared to that of pristine CdS NWs.” has been added.
 17. On line 21 of page 16, “After the growth of III-V NWs, zone 1 is heated to the growth temperature of chalcogenide semiconductors immediately, and the holding time is adjusted to realize the controllable growth of the amorphous shell. It is worth pointing out that zone 2 needs cooled naturally to the same temperature as zone 1.” has been added.
 18. On line 27 of page 16, “The growth details of core-shell heterostructure NWs can be found in Table S2.” has been added.
 19. On line 11 of page 17, “The band structures of the as-constructed core-shell heterostructure NWs are determined by the UPS (ESCALAB XI+, ThermoFisher).” has been revised.

Other Changes List in the Revised Manuscript

The section of “References” has been revised as below.

1. del Alamo, J. A. Nanometre-scale electronics with III-V compound semiconductors. *Nature* **479**, 317-323 (2011).
2. Kammhuber, J. *et al.* Conductance through a helical state in an Indium antimonide nanowire. *Nat. Commun.* **8**, 478 (2017).
3. Gibson, S. J. *et al.* Tapered InP nanowire arrays for efficient broadband high-speed single-photon detection. *Nat. Nanotechnol.* **14**, 473-479 (2019).
4. Mauthe, S. *et al.* High-speed III-V nanowire photodetector monolithically integrated on Si. *Nat. Commun.* **11**, 4565 (2020).
5. Peng, K. *et al.* Three-dimensional cross-nanowire networks recover full terahertz state. *Science* **368**, 510-513 (2020).
6. Nela, L. *et al.* Multi-channel nanowire devices for efficient power conversion. *Nat. Electron.* **4**, 284-290 (2021).
7. Yi, R. *et al.* Self-frequency-conversion nanowire lasers. *Light Sci. Appl.* **11**, 120 (2022).
8. Wang, J.-Y. *et al.* Supercurrent parity meter in a nanowire Cooper pair transistor. *Sci. Adv.* **8**, eabm9896 (2022).
9. Wang, D. *et al.* Observation of polarity-switchable photoconductivity in III-nitride/MoS_x core-shell nanowires. *Light Sci. Appl.* **11**, 227 (2022).
10. Mayer, B. *et al.* Lasing from individual GaAs-AlGaAs core-shell nanowires up to room temperature. *Nat. Commun.* **4**, 2931 (2013).
11. Balaghig, L. *et al.* Widely tunable GaAs bandgap via strain engineering in core/shell nanowires with large lattice mismatch. *Nat. Commun.* **10**, 2793 (2019).
12. Pendharkar, M. *et al.* Parity-preserving and magnetic field-resilient superconductivity in InSb nanowires with Sn shells. *Science* **372**, 508-511 (2021).
13. Oksenberg, E., Marti-Sanchez, S., Popovitz-Biro, R., Arbiol, J. & Joselevich, E. Surface-guided core-shell ZnSe@ZnTe nanowires as radial p-n heterojunctions with photovoltaic behavior. *ACS Nano* **11**, 6155-6166 (2017).
14. Wang, H. *et al.* Slowing hot-electron relaxation in mix-phase nanowires for hot-carrier photovoltaics. *Nano Lett.* **21**, 7761-7768 (2021).
15. Zhang, J. *et al.* Enhance the responsivity and response speed of self-powered ultraviolet photodetector by GaN/CsPbBr₃ core-shell nanowire heterojunction and hydrogel. *Nano Energy* **100**, 107437 (2022).
16. Treu, J. *et al.* Lattice-matched InGaAs-InAlAs core-shell nanowires with improved luminescence and photoresponse properties. *Nano Lett.* **15**, 3533-3540 (2015).
17. Lv, Q. *et al.* Lattice-mismatch-free growth of organic heterostructure nanowires from cocrystals to alloys. *Nat. Commun.* **13**, 3099 (2022).
18. Haapamaki, C. M., Baugh, J. & LaPierre, R. R. Critical shell thickness for InAs-Al_xIn_{1-x}As(P) core-shell nanowires. *J. Appl. Phys.* **112**, 124305 (2012).
19. Li, X. *et al.* Contactless optical characterization of carrier dynamics in free-standing InAs-InAlAs core-shell nanowires on silicon. *Nano Lett.* **19**, 990-996 (2019).
20. Arif, O. *et al.* Growth and strain relaxation mechanisms of InAs/InP/GaAsSb core-dual-shell nanowires. *Cryst. Growth Des.* **20**, 1088-1096 (2020).
21. Li, L. *et al.* Integrated flexible chalcogenide glass photonic devices. *Nat. Photonics* **8**, 643-649 (2014).
22. Lin, H. *et al.* Chalcogenide glass-on-graphene photonics. *Nat. Photonics* **11**, 798-805 (2017).
23. Jia, S. *et al.* Ultrahigh drive current and large selectivity in GeS selector. *Nat. Commun.* **11**, 4636 (2020).
24. Ohno, T. Passivation of GaAs(001) surfaces by chalcogen atoms (S, Se and Te). *Surf. Sci.* **255**, 229-236 (1991).
25. Sun, M. H. *et al.* Removal of surface states and recovery of band-edge emission in InAs nanowires through surface passivation. *Nano Lett.* **12**, 3378-3384 (2012).

26. Ohtake, A., Goto, S. & Nakamura, J. Atomic structure and passivated nature of the Se-treated GaAs(111) B surface. *Sci. Rep.* **8**, 1220 (2018).
27. Yang, Z.-x. *et al.* Surfactant-assisted chemical vapour deposition of high-performance small-diameter GaSb nanowires. *Nat. Commun.* **5**, 5249 (2014).
28. Sun, J. *et al.* Toward Unusual-High Hole Mobility of p-Channel Field-Effect-Transistors. *Small* **17**, 2102323 (2021).
29. Ren, Z. *et al.* Short-wave near-infrared polarization sensitive photodetector based on GaSb nanowire. *IEEE Electron Device Lett.* **42**, 549-552 (2021).
30. Liu, D. *et al.* Schottky-Contacted High-Performance GaSb Nanowires Photodetectors Enabled by Lead-Free All-Inorganic Perovskites Decoration. *Small* **18**, 2200415 (2022).
31. Yang, Z.-x. *et al.* Complementary metal oxide semiconductor-compatible, high-mobility, < 111 >-oriented GaSb nanowires enabled by vapor-solid-solid chemical vapor deposition. *ACS Nano* **11**, 4237-4246 (2017).
32. Takebe, H., Maeda, H. & Morinaga, K. Compositional variation in the structure of Ge-S glasses. *J. Non-Cryst. Solids* **291**, 14-24 (2001).
33. Sakaguchi, Y., Hanashima, T., Ohara, K., Simon, A.-A. A. & Mitkova, M. Structural transformation in $\text{Ge}_x\text{S}_{100-x}$ ($10 \leq x \leq 40$) network glasses: Structural varieties in short-range, medium-range, and nanoscopic scale. *Phys. Rev. Mater.* **3**, 035601 (2019).
34. Yin, Y. *et al.* Substrate-free chemical vapor deposition of large-scale III-V nanowires for high-performance transistors and broad-spectrum photodetectors. *Adv. Optical Mater.* **10**, 2102291 (2022).
35. Chen, H. *et al.* Termination of Ge surfaces with ultrathin GeS and GeS_2 layers via solid-state sulfurization. *Phys. Chem. Chem. Phys.* **19**, 32473-32480 (2017).
36. Zhao, S. *et al.* In situ growth of GeS nanowires with sulfur-rich shell for featured negative photoconductivity. *J. Phys. Chem. Lett.* **12**, 3046-3052 (2021).
37. Lauhon, L. J., Gudiksen, M. S., Wang, C. L. & Lieber, C. M. Epitaxial core-shell and core-multishell nanowire heterostructures. *Nature* **420**, 57-61 (2002).
38. Sa, Z. *et al.* Toward high bias-stress stability p-type GaSb nanowire field-effect-transistor for gate-controlled near-infrared photodetection and photocommunication. *Adv. Funct. Mater.* DOI: 10.1002/adfm.2023040 (2023).
39. Chen, Y. *et al.* Unipolar barrier photodetectors based on van der Waals heterostructures. *Nat. Electron.* **4**, 357-363 (2021).
40. Wang, D. *et al.* Bidirectional photocurrent in p-n heterojunction nanowires. *Nat. Electron.* **4**, 645-652 (2021).
41. Hwang, A. *et al.* Visible and infrared dual-band imaging via Ge/MoS₂ van der Waals heterostructure. *Sci. Adv* **7**, eabj2521 (2021).
42. Sun, J. *et al.* Stoichiometric effect on electrical and near-infrared photodetection properties of full-composition-range $\text{GaAs}_{1-x}\text{Sb}_x$ nanowires. *Nano Res.* **14**, 3961-3968 (2021).
43. Yang, Y. *et al.* Hot carrier trapping induced negative photoconductance in InAs nanowires toward novel nonvolatile memory. *Nano Lett.* **15**, 5875-5882 (2015).
44. Fang, H. *et al.* Visible Light-assisted high-performance mid-infrared photodetectors based on single InAs nanowire. *Nano Lett.* **16**, 6416-6424 (2016).
45. Fang, H. & Hu, W. Photogating in low dimensional photodetectors. *Adv. Sci.* **4**, 1700323 (2017).
46. Bonilla, R. S. & Wilshaw, P. R. A technique for field effect surface passivation for silicon solar cells. *Appl. Phys. Lett.* **104**, 232903 (2014).
47. Mallorqui, A. D. *et al.* Field-effect passivation on silicon nanowire solar cells. *Nano Res.* **8**, 673-681 (2015).
48. Zheng, Z., Zu, X., Zhang, Y. & Zhou, W. Rational design of type-II nano-heterojunctions for nanoscale optoelectronics. *Mater. Today Phys.* **15**, 100262 (2020).
49. Fan, Z. *et al.* Wafer-scale assembly of highly ordered semiconductor nanowire arrays by contact printing. *Nano Lett.* **8**, 20-25 (2008).
50. Dai, X. *et al.* GaAs/AlGaAs nanowire photodetector. *Nano Lett.* **14**, 2688-2693 (2014).

-
51. Wang, X. *et al.* Vis-IR wide-spectrum photodetector at room temperature based on p-n junction-type GaAs_{1-x}Sb_x/InAs core-shell nanowire. *ACS Appl. Mater. Interfaces* **11**, 38973-38981 (2019).
 52. Li, H. *et al.* Novel type-II InAs/AlSb core-shell nanowires and their enhanced negative photocurrent for efficient photodetection. *Adv. Funct. Mater.* **28**, 1705382 (2018).
 53. Gallo, E. M. *et al.* Picosecond response times in GaAs/AlGaAs core/shell nanowire-based photodetectors. *Appl. Phys. Lett.* **98**, 241113 (2011).
 54. Zhou, C. *et al.* Self-assembly growth of In-rich InGaAs core-shell structured nanowires with remarkable near-infrared photoresponsivity. *Nano Lett.* **17**, 7824-7830 (2017).

Changes List in the Revised Supporting Information

1. Fig. S3, including the captions, has been added as follows.

Fig. S3 | TEM images of as-constructed GaSb/GeS core-shell heterostructure NWs. All the scale bars are 50 nm.

2. Fig. S5, including the captions and corresponding discussion has been added as follows.

Fig. S5 | Band alignments of as-constructed core-shell heterostructure NWs. a-e UPS of GaSb/GeS core-shell heterostructure NWs, pristine GaSb NWs, pristine GaAs NWs, pristine InGaAs NWs, and GaSb/GeSe core-shell heterostructure NWs, respectively. The solid black lines mark the baselines and the tangents of the curves. The intersections of the tangents with the baselines indicate the edges of UPS. f-i, Type-I band alignment of GaSb/GeS heterostructure, Type-II band alignments of GaAs/GeS heterostructure, InGaAs/GeS heterostructure, and GaSb/GeSe heterostructure.

Ultraviolet photoelectron spectroscopy (UPS, ESCALAB XI+, ThermoFisher) is adopted to evaluate the band structures of as-constructed core-shell heterostructure NWs. The band structures are deduced from UPS of pristine GaSb NWs, GaSb/GeS core-shell heterostructure NWs, pristine GaAs NWs, pristine InGaAs NWs, and GaSb/GeSe core-shell heterostructure NWs. According to the linear intersection method, the valence band (E_V) values of GaSb, GeS, GaAs, InGaAs, and GeSe are calculated as - 4.73 eV, - 5.27 eV, - 5.81 eV, - 5.18 eV, and - 5.73 eV, respectively, by subtracting the width of He I UPS from the excitation energy of 21.21 eV. Meanwhile, the work functions of GaSb, GeS, GaAs, InGaAs, and GeSe are calculated as - 4.46 eV, - 4.73 eV, - 5.11 eV, - 4.40 eV, and - 5.12 eV, respectively, by adding E_V to the second electron cutoff energy. According to the bandgap values of GaSb (0.70 eV), GeS (1.50 eV), GaAs (1.40 eV), InGaAs (1.13 eV), and GeSe (0.97 eV) reported in the literatures¹⁻⁴, the conduction band (E_C) values of GaSb, GeS, GaAs, InGaAs, and GeSe are calculated as - 4.03 eV, - 3.77 eV, - 4.41 eV, - 4.05 eV and - 4.76 eV, respectively. In this case, the band alignment diagrams of as-constructed GaSb/GeS, GaAs/GeS, InGaAs/GeS, and GaSb/GeSe core-shell heterostructure NWs are drawn approximated. Obviously, the heterostructures with expected heterostructure type, such as type-I of GaSb/GeS and type-II of GaAs/GeS, InGaAs/GeS, and GaSb/GeSe are constructed successfully.

3. Fig. S7, including the captions and corresponding discussion, has been added as follows.

Fig. S7 | Electrical properties of GaSb/GeS core-shell heterostructure NWs. a, I-V curves of pristine GaSb NW and GaSb/GeS core-shell heterostructure NWs with different shell thicknesses. **b,** Schematic of GaSb/GeS core-shell heterostructure NW MSM photodetector. **c,** Hole carriers transfer process in GaSb/GeS core-shell NWs under a large bias voltage.

The electrical properties of the as-constructed GaSb/GeS core-shell heterostructure NWs are presented in Fig. S7. For the pristine GaSb NW, the I_{DS} of the as-fabricated MSM photodetector changes linearly with V_{DS} , demonstrating the typical Ohmic contacts between Ni electrodes and GaSb NW, as shown in Fig. R2a. With a V_{DS} of 3V, the I_{DS} is around 3.96 μ A. This result is in line with the results of previous literature^{5,6}. For the GaSb/GeS core-shell heterostructure NWs, the I_{DS} of as-fabricated MSM photodetectors also changes linearly under a small bias voltage, appearing as saturated I_{DS} under a large bias voltage. With the V_{DS} around 1V, the saturated I_{DS} are around 304, 217, and 12 nA for thin, medium, and thick shell NWs, respectively. Although the I_{DS} are smaller than that of pristine GaSb NW MSM photodetector, they are

unlikely the leakages of amorphous GeS films, which can be deduced from the observation of saturated I_{DS} in the I-V curves. Furthermore, based on the infrared photodetection behaviors of as-constructed GaSb/GeS core-shell heterostructure NWs and InGaAs/GeS core-shell heterostructure NWs, the cores of GaSb and InGaAs act as the main conductive channels. In this case, the transfer path of hole carriers is Ni electrode to GeS shell to GaSb core to GeS shell to Ni electrode, as illustrated in Fig. R2b. The thicker GeS shell gives the larger resistance of the shell, resulting in the smaller I_{DS} . Under a large voltage bias, the mechanism of saturated I_{DS} is presented in Fig. R2c. Taking the positive bias as an example. Due to the small valence band offset (ΔE_v) between the GeS shell and GaSb core, the hole carriers in the GaSb core will flow across the reverse-biased heterojunction easily under small bias voltage, and the I-V curve presents linear characteristics. However, when the bias voltage increases further, the valence band offset of the reverse bias heterojunction will increase to ($\Delta E_v + e\Delta V$), which hinders the transfer of hole carriers from the GaSb core to GeS shell, resulting in the saturation of I_{DS} .

4. Fig. S9, including the captions and corresponding discussion, has been added as follows.

Fig. S9 | Visible light-assisted infrared photodetection performance of GaSb/GeS core-shell heterostructure NWs. **a-b**, The visible light-assisted infrared photodetection performance of NWs with a medium shell, where 520 nm and 635 nm light act as assisted light, respectively. **c-d**, The visible light-assisted infrared photodetection performance of NWs with a thick shell, where 520 nm and 635 nm light act as assisted light, respectively.

Based on the mechanism of visible light-assisted behavior, both 520 nm and 635 nm lights also can improve the infrared photodetection performance of photodetector fabricated by GaSb/GeS core-shell heterostructure NWs with an appropriate shell thickness. As shown in Fig. S9a, with a medium shell, when assisted light of 520 nm is on, the dark current of the infrared photodetector decreases from 269 nA to 187

nA. At the same time, the photocurrent increases from 10 nA to 27 nA. When auxiliary light of 635 nm is on (Fig. S9b), the dark current of the infrared photodetector decreases from 266 nA to 216 nA, and the photocurrent increases from 11 nA to 22 nA. For the photodetector with a thick shell, with the assistance of visible lights of 520 nm and 635 nm, the dark currents of the photodetectors for infrared light are reduced, as shown in Fig. S9c and S9d. But barely any improvement of infrared photodetection currents is observed, similar to 405 nm assisted light (Fig. S14b).

5. Fig. S11, including the captions and corresponding discussion, has been added as follows.

Fig. S11 | Construction of GaSb/GeS core-shell heterostructure NWs with ultra-thin GeS shells and their photodetection behaviors. **a**, HRTEM image of GaSb/GeS core-shell heterostructure NW with ultra-thin shell. **b**, EDS elemental mapping images of Ga, Sb, Al, O. The inset is the corresponding scanning transmission electron microscopy (STEM) image. All the scale bars are 20 nm. **c**, Broad-spectrum photodetection behaviors of pristine GaSb NWs and GaSb/GeS core-shell heterostructure NWs with ultra-thin shell. The laser intensities from 405 nm to 785 nm and 850 nm to 1550 nm are $0.05 \text{ mW}\cdot\text{mm}^{-2}$ and $6.0 \text{ mW}\cdot\text{mm}^{-2}$, respectively.

The ultra-thin GeS shell also grows on the surfaces of GaSb NWs for studying the passivation effect. From the HRTEM image of Fig. S11a, with a growth time of 1 s, the shell of as-constructed GaSb/GeS core-shell heterostructure NWs is around 2 nm. As shown in the EDS mapping images of Fig S11b, Ga and Sb dominate the NW core. At the same time, Ge and S dominate the shell. This finding is in line with the result of GaSb/GeS core-shell heterostructure NW with a thicker shell, as shown in Fig. 1. From the broad-spectrum photodetection behavior of Fig. S11c, it is found that GaSb/GeS core-shell heterostructure NW exhibits larger photocurrent compared to that of pristine GaSb NW, which is attributed to the surface passivation effect of ultra-thin GeS shell. At the same time, due to the negative photoresponse caused by the GeS shell, a reduced photocurrent is also observed at the near ultraviolet waveband of 405 nm. In short,

the amorphous chalcogenide shells not only overcome the lattice mismatch, but also passivate the surface charge trappings of III-V NWs.

6. Fig. S12, including the captions and corresponding discussion, has been added as follows.

Fig. S12 | Construction of GaSb/Al₂O₃ core-shell NWs and their photodetection behaviors. **a**, HRTEM image of GaSb/Al₂O₃ core-shell NW. **b**, EDS elemental mapping images of Ga, Sb, Al, O. All the scale bars are 20 nm. **c**, **d**, Broad-spectrum photodetection behaviors of pristine GaSb NWs and GaSb/Al₂O₃ core-shell NWs with 2 nm and 5 nm Al₂O₃ shell, respectively. The laser intensities from 405 nm to 785 nm and 850 nm to 1550 nm are 0.05 mW·mm⁻² and 6.0 mW·mm⁻², respectively. **e**, **f**, Infrared photodetection behaviors of GaSb/Al₂O₃ core-shell NWs with 2 nm and 5 nm Al₂O₃ shells under the illuminations of visible light, respectively. The laser intensities of 405 nm and 1550 nm are 0.05 mW·mm⁻² and 5.0 mW·mm⁻², respectively.

Beyond amorphous GeS, the larger bandgap Al₂O₃ is also attempted to passivate the surface charge trappings of GaSb NWs, as shown in Fig. S12. The Al₂O₃ shells grow on the surfaces of GaSb NWs by atomic layer deposition method. The HRTEM image and EDS mapping images of Fig. S12a-b show that GaSb/Al₂O₃ core-shell heterostructure NW with a shell of 5 nm is successfully constructed. Ga and Sb

dominate the NW core. On the other hand, Al and O dominate the shell. From the broad-spectrum photodetection behaviors of Fig. S12c-d, it is found that GaSb NWs with the Al₂O₃ shells of 2 nm and 5 nm both exhibit larger photocurrents compared to the pristine GaSb NW, which is attributed to the surface passivation effect of Al₂O₃ shells. Furthermore, the visible light-assisted infrared photodetection behaviors are also studied in Fig. S12e-f. When the visible light is on, the infrared photodetection currents can be distinguished hardly. This finding can be attributed to the fact that a large number of carriers generated by visible light act as background carriers for infrared photodetection, which leads to the serious recombination of photogenerated carriers (generated by 1500 nm laser)^{7,8}. The results show that the epitaxial larger bandgap Al₂O₃ shells can passivate the surface charge trappings of GaSb NWs effectively. Compared to the as-constructed III-V/chalcogenide core-shell heterostructure NWs, wavelength-dependent bi-directional photodetection behavior, visible light-assisted infrared photodetection behavior, and faster response times are not observed in GaSb/Al₂O₃ core-shell heterostructure NWs.

7. Fig. S13, including the captions and corresponding discussion, has been added as follows.

Fig. S13 | Stability of as-fabricated GaSb/GeS core-shell heterostructure NW photodetector. a, Broad-spectrum photodetection behavior of GaSb/GeS core-shell heterostructure NW after being stored in an atmospheric environment for 10 days and 30 days. The laser intensities from 405 nm to 785 nm and 850 nm to 1550 nm are 0.05 mW·mm⁻² and 6.0 mW·mm⁻², respectively. **b,** Visible light-assisted infrared photodetection performance of GaSb/GeS core-shell heterostructure NW after being stored in an atmospheric environment for 30 days.

As presented in Fig. S13a, the as-fabricated photodetector still exhibits stable wavelength-dependent bi-directional photodetection behavior, which displays a negative photoresponse in the wavelength of 405-785 nm and a positive photoresponse in the wavelength of 850-1550 nm after being stored in an atmospheric environment for 10 days and 30 days. The photocurrent attenuations are less than 10% and 20% for 10 days and 30 days, respectively. As shown in Fig. S13b, the as-fabricated photodetector also shows the visible light-assisted infrared photodetection behavior after being stored in an atmospheric environment for 30 days. It is found that when the assisted light of 405 nm is on, the dark current of the infrared photodetector is significantly suppressed, and the photocurrent increases obviously. In short, benefitting the as-constructed core-shell nanostructure, the as-fabricated photodetector exhibits stable, repeatable, and robust photodetection performance, which promises the application in further optoelectronic devices.

8. Fig. S14, including the captions and corresponding discussion, has been added as follows.

Fig. S14. The visible light-assisted infrared photodetection performance of NWs with (a) a thin shell and (b) a thick shell.

The shell thickness also plays an important role in visible light-assisted behavior. As shown in Fig. S14a, when the visible light (405 nm) is on, the infrared photodetection currents are suppressed in the thin shell core-shell NWs photodetector. This suppression is because the absorption of visible light is dominated by the GaSb core in the thin shell core-shell NWs. In this case, the photogenerated carriers generated by 1500 nm laser are annihilated in the large number of carriers generated by visible light, resulting in suppressed photocurrents. With a medium shell, the visible light is mainly absorbed by the GeS shell. When the visible light is on, the dark current of the photodetector for infrared light is reduced due to the photogenerated electrons in the GeS shell recombining with the free holes in the GaSb core, as shown in Fig. 4e. Meanwhile, the trapped photogenerated holes in the GeS shell lead to the field-effect passivation^{9,10}, which would inhibit the recombination of the infrared light excited electron-hole pairs in the GaSb core, ultimately leading to the improvement of infrared photodetection currents. With a thick shell, the dark current is also decreased with visible light, as shown in Fig. S14b. But barely any improvement of infrared photodetection current is observed, possibly due to the thick shell weakening the field-effect passivation effect.

9. Fig. S17, including the captions and corresponding discussion, has been added as follows.

Fig. S17 | Lattice-mismatch-free construction of CdS/GeS core-shell heterostructure NWs and their photodetection behaviors. **a**, HRTEM image of CdS/GeS core-shell heterostructure NW. **b**, EDS elemental mapping images of Cd, Ge, S. All the scale bars are 20 nm. **c**, XRD patterns of CdS NWs and as-constructed core-shell heterostructure NWs. **d**, Photodetection behaviors of pristine CdS NW and CdS/GeS core-shell heterostructure NW. **e**, Schematic for type-II CdS/GeS core-shell NWs photodetector.

In this work, a versatile strategy is exploited for the lattice-mismatch-free construction of III-V/chalcogenide core-shell heterostructure NWs by simply utilizing the surfactant and amorphous natures of chalcogenide semiconductors. Theoretically, this approach also can be used for lattice-mismatch-free construction of chalcogenide semiconductors core-shell heterostructure NWs. As shown in Fig. S17, the as-expected CdS/GeS core-shell heterostructure NW and the rational band alignment are successfully constructed. From the HRTEM image (Fig. S17a), the as-prepared NW shows apparent contrast between the core and shell. It is also obvious that the shell conformally wraps around the core. The clear lattice fringes are observed with the lattice spacings of 0.65 and 0.36 nm, indicating the good crystallinity of core NW. Noteworthy, no obvious crystal lattice fringes on the surface indicate the shell is amorphous. The elemental compositions of the core and shell are checked by EDS elemental mappings in Fig. S17b. The distribution of element Cd mainly concentrates in the core region, while elements Ge and S are observed in the whole NW, inferring that CdS/GeS core-shell heterostructure NW is successfully constructed. In the end, X-ray diffraction (XRD) further verifies the amorphous shell in Fig. S17c. Furthermore, the as-constructed CdS/GeS core-shell heterostructure NWs exhibit excellent photodetection performance with an extended spectrum detection range and much larger photocurrent compared to that of pristine CdS NWs, as shown in Fig. S17d. The improved photodetection performance is attributed to the successful

construction of type-II p-n core-shell heterostructure, as illustrated in Fig. S17e. Obviously, the developed lattice-mismatch-free construction strategy is efficient and versatile, not only for III-V NWs but also for chalcogenides.

10. Table S1 has been added as follows.

Table S1. Photodetection performances comparison of III-V core-shell heterostructure NWs.

Core-shell NWs	Growth method	Heterostructure type	Photoresponse behavior	R (A/W)	D* (Jones)	Response time (t _r /t _d)	Ref.
GaAs/AlGaAs	MOCVD	Type-I	Positive	0.57 (@855 nm)	7.2×10 ¹⁰		11
GaAs _{1-x} Sb _x /InAs	MBE	Type- II p-n	Positive	0.12 (@1310 nm)	-	0.45 ms/0.86 ms (633nm)	12
InAs/AlSb	MBE	Type-II p-n	Negative	-	-	-	13
GaAs/AlGaAs	MOVPE	Type-I	Positive	10 ⁻⁴ (@800 nm)	-	5 ps	14
In-Rich InGaAs	MBE	-	Positive	5.75 (@1550 nm)	-	-	15
GaSb/GeS	CVD	Type-I p-p	Wavelength dependent bi-directional	400 (@1550 nm)	9.8×10 ¹⁰	12 ms/8 ms (1550 nm)	This work
InGaAs/GeS	CVD	Type-II p-n	Positive	61.2 (@850 nm)	6.8×10 ¹¹	3.3 ms/2.9 ms (850 nm)	This work

11. Table S2 has been added as follows.

Table S2. Growth details of core-shell heterostructure NWs.

Core-shell NWs	Catalyst (nm)	Growth of core NWs					Growth of shells			
		Source (g)	Sulfur (g)	S.T. (°C)	G.T. (°C)	Carrier gas (sccm)	Source (g)	S.T. (°C)	G.T. (°C)	Carrier gas (sccm)
GaSb/GeS	Au/1.0	GaSb/0.4	0.4	750	560	H ₂ /200	GeS/0.1	500	320	H ₂ /200
GaAs/GeS	Au/1.0	GaAs/0.4	-	800	600	H ₂ /200	GeS/0.1	500	320	H ₂ /200
InGaAs/GeS	Ni/1.0	(InAs:GaAs =1:1)/0.4	-	810	610	H ₂ /200	GeS/0.1	500	320	H ₂ /200
GaSb/GeSe	Au/1.0	GaSb/0.4	0.4	750	560	H ₂ /200	GeSe/0.1	550	320	H ₂ /200
CdS/GeS	Au/1.0	CdS/0.1	-	750	540	Ar/50	GeS/0.1	500	320	H ₂ /200
GaSb/GaAs	Au/1.0	GaSb/0.4	0.4	750	560	H ₂ /200	GaAs/0.4	750	560	H ₂ /200

Note: S.T.: Source temperature; G.T.: Growth temperature; Growth substrate: Si/SiO₂; Pressure of CVD system: 6×10⁻³ Torr; Position of source and growth substrate: source powders of shell semiconductors (such as GeS, GeSe, and GaAs) are placed in upstream zone 1, source powders of core semiconductors (such as GaSb, GaAs, InGaAs, and CdS) are placed in midstream zone 2, the growth substrate is placed in the downstream zone 3, and sulfur powders are placed between zone 2 and 3.

12. The section of “References” has been revised as below.

- Gobeli, G. W. & Allen, F. G. Photoelectric properties of cleaved GaAs GaSb InAs and InSb surfaces-comparison with Si and Ge. *Phys. Rev.* **137**, A245-A254 (1965).

2. Jia, S. *et al.* Ultrahigh drive current and large selectivity in GeS selector. *Nat. Commun.* **11**, 4636 (2020).
3. Zhang, D. & Li, Z. InP/ZnS quantum dots functionalized AlGaAs/InGaAs open gate high electron mobility transistor. *J. Mater. Sci.: Mater. Electron.* **29**, 10663-10668, (2018).
4. Ahn, H.-W. *et al.* Effect of density of localized states on the ovonic threshold switching characteristics of the amorphous GeSe films. *Appl. Phys. Lett.* **103**, 042908 (2013).
5. Yang, Z.-x. *et al.* Surfactant-assisted chemical vapour deposition of high-performance small-diameter GaSb nanowires. *Nat. Commun.* **5**, 5249 (2014).
6. Sun, J. *et al.* Ultrahigh hole mobility of Sn-catalyzed GaSb nanowires for high speed infrared photodetectors. *Nano Lett.* **19**, 5920-5929, (2019).
7. Lee, H. K. H. *et al.* The role of fullerenes in the environmental stability of polymer: fullerene solar cells. *Energy Environ. Sci.* **11**, 417-428 (2018).
8. Caprioglio, P. *et al.* On the origin of the ideality factor in perovskite solar cells. *Adv. Energy Mater.* **10**, 2000502 (2020).
9. Bonilla, R. S. & Wilshaw, P. R. A technique for field effect surface passivation for silicon solar cells. *Appl. Phys. Lett.* **104**, 232903 (2014).
10. Mallorqui, A. D. *et al.* Field-effect passivation on silicon nanowire solar cells. *Nano Res.* **8**, 673-681 (2015).
11. Lu, Z. *et al.* Ultrahigh speed and broadband few-layer MoTe₂/Si 2D-3D heterojunction-based photodiodes fabricated by pulsed laser deposition. *Adv. Funct. Mater.* **30**, 1907951 (2020).
12. Dai, X. *et al.* GaAs/AlGaAs nanowire photodetector. *Nano Lett.* **14**, 2688-2693 (2014).
13. Wang, X. *et al.* Vis-IR wide-spectrum photodetector at room temperature based on p-n junction-type GaAs_{1-x}Sb_x/InAs core-shell nanowire. *ACS Appl. Mater. Interfaces* **11**, 38973-38981 (2019).
14. Li, H. *et al.* Novel type-II InAs/AlSb core-shell nanowires and their enhanced negative photocurrent for efficient photodetection. *Adv. Funct. Mater.* **28**, 1705382 (2018).
15. Gallo, E. M. *et al.* Picosecond response times in GaAs/AlGaAs core/shell nanowire-based photodetectors. *Appl. Phys. Lett.* **98**, 241113 (2011).
16. Zhou, C. *et al.* Self-assembly growth of In-rich InGaAs core-shell structured nanowires with remarkable near-infrared photoresponsivity. *Nano Lett.* **17**, 7824-7830 (2017).

REVIEWER COMMENTS

Reviewer #1 (Remarks to the Author):

The authors to the manuscript have done a very thorough review of their manuscript and updated with very adequate information both in the reply to the reviewers and with useful modifications to the manuscript itself. New experimental data are included from new experiments. It is very much appreciated!

Although the manuscript is very well written and contains much useful information - in particular for the material section and the optical and XPS characterization - I do not think it should be published in its present form. From the IV curves and the band diagram provided in figure S7 (thanks for the additional information on the contact behaviour) it is clear that the hole injection across the shell structure (or alternatively tunnelling or defect assisted tunneling across the layer at the contact) is limiting the current in the devices. Much of the photoresponse will thus be associated with the transport at the contacts. I think this seriously will change the discussion on the device physics in these nanowires although the nice measured data will stand by themselves. Therefore I suggest that the manuscript be rejected at this point.

Reviewer #2 (Remarks to the Author):

Authors have addressed all the concerns I have raised and I am satisfied with their additional experiments and data analysis. I have no additional comments and suggestions. I am happy to accept the manuscript for publication.

Reviewer #3 (Remarks to the Author):

In the revision of the manuscript the authors have addressed the points raised by the referees.

The revised manuscript can be now accepted for publication.

Response to the Reviewers' Comments on Manuscript NCOMMS-23-10752A

We appreciate the reviewers for considering our manuscript and providing valuable comments. Accordingly, changes have been made in the manuscript, highlighted in red color. Below is our response to the reviewers' comments.

Response to the Reviewers' comments:

Reviewer 1:

Reviewer's comments:

The authors to the manuscript have done a very thorough review of their manuscript and updated with very adequate information both in the reply to the reviewers and with useful modifications to the manuscript itself. New experimental data are included from new experiments. It is very much appreciated!

Although the manuscript is very well written and contains much useful information - in particular for the material section and the optical and XPS characterization - I do not think it should be published in its present form. From the IV curves and the band diagram provided in figure S7 (thanks for the additional information on the contact behaviour) it is clear that the hole injection across the shell structure (or alternatively tunnelling or defect assisted tunneling across the layer at the contact) is limiting the current in the devices. Much of the photoresponse will thus be associated with the transport at the contacts. I think this seriously will change the discussion on the device physics in these nanowires although the nice measured data will stand by themselves. Therefore I suggest that the manuscript be rejected at this point.

Response:

We thank reviewer #1 for the rapid and positive response to our manuscript. We also agree with the viewpoint of reviewer #1. Since both the hole injection and collection pass across the GeS shell, the hole transport at the contacts is important, requiring a detailed study. Accordingly, we checked the device behaviors carefully, and a more detailed discussion about the hole transport at the contacts has been added in the revised manuscript and supplementary information.

Discussion on the current limiting role of GeS shell.

Fig. R1 shows the electrical properties of as-constructed GaSb/GeS core-shell heterostructure NWs. Based on the I-V curves of Fig. R1a, the I_{DS} of the pristine GaSb NW MSM photodetector changes linearly with V_{DS} , demonstrating the typical ohmic-like contacts between Ni electrodes and GaSb NW. With a V_{DS} of 3V, the I_{DS} is 3.96 μ A. For the GaSb/GeS core-shell heterostructure NWs MSM photodetectors, the I_{DS} is significantly reduced compared to the pristine GaSb NW MSM photodetector. At the same time, the

saturated I_{DS} is observed under large bias voltages. The saturated I_{DS} decreases from 304 to 217 and 12 nA for thin, medium, and thick shell NWs, respectively. In short, with the increase of shell thickness, the I_{DS} of as-studied MSM photodetectors decreases, indicating that the GaSb cores act as the main conductive channels (as shown in Fig. R1b), while the GeS shells limit the current.

Fig. R1 | Electrical properties of GaSb/GeS core-shell heterostructure NWs. a, I-V curves of the pristine GaSb NW and GaSb/GeS core-shell heterostructure NWs with different shell thicknesses. **b,** Schematic of NW MSM photodetector. **c,** Hole transport in the GaSb/GeS core-shell heterostructure NW.

To further illustrate the current limiting mechanism, a band structure model is depicted in Fig. R1c to demonstrate the hole transport process at the interface of GaSb and GeS. Under a forward bias voltage, the holes are injected from the drain electrode to the main conductive channel of the GaSb core and are collected at the source electrode, as shown in Fig. R1b. The holes pass across the GeS shell during both the injection and collection processes. As shown in Fig. R1c, a forward bias heterojunction and a reverse bias heterojunction are formed at the injection and collection regions, which block the hole transport and result in decreased currents compared to the pristine GaSb NW MSM photodetector. At thermal equilibrium, the barrier of the forward bias heterojunction (due to the Fermi level coincident) is $q\psi_b$ (q is the electronic charge, and ψ_b is the electrostatic potential), and the barrier of the reverse bias heterojunction (due to the valence band offset) is ΔE_V (E_V is the valence band edge) (*Light Sci. Appl.* 2019, 8, 106; *Nat. Commun.* 2019, 10, 4663). Obviously, with the increase of bias voltage, the barrier of the forward heterojunction of $q\psi_b$ at injection region decreases, which will benefit the hole transport, resulting in the increase of I_{DS} . At the same time, the barrier of the reverse heterojunction at the collection region keeps ΔE_V . In this case, the saturated I_{DS} is observed at a large bias voltage. It is worth mentioning that in addition to the classical thermionic emission, tunneling or defect-assisted tunneling also plays a role in the holes injection and collection processes, as proposed by reviewer #1 (*Nat. Phys.* 2016, 12, 455; *Adv. Sci.* 2020, 7, 1902751). With a thin shell, the barrier is possibly thin and low, benefiting the hole tunneling process. As a result, with the increase in the shell thickness, the I_{DS} decreases.

In this case, Fig. S7 has been revised as Fig. R1, and the discussion of "As shown in the I-V curves of Fig. S7, with the increase of shell thickness, the I_{DS} of as-studied MSM photodetectors decreases, indicating that

the GaSb cores act as the main conductive channels, while the GeS shells limit the current owing to the formation of heterojunctions at the contacts. It is worth mentioning that in addition to the classical thermionic emission, tunneling or defect-assisted tunneling also plays a role in the hole injection and collection processes^{42,43}. With a thin shell, the barrier is possibly thin and low, benefiting the hole tunneling. As a result, with the increase in the shell thickness, the I_{DS} decreases. Because both the hole injection and collection processes pass across the GeS shell, it also plays an important role in the transport of photogenerated carriers at the contacts, and the details will be discussed later." has been added on line 26 of page 10 in the revised manuscript.

At the same time, the discussion of Fig. S7 has also been revised as "Fig. S7 shows the electrical properties of as-constructed GaSb/GeS core-shell heterostructure NWs. Based on the I-V curves of Fig. S7a, the I_{DS} of the pristine GaSb NW MSM photodetector changes linearly with V_{DS} , demonstrating the typical ohmic-like contacts between Ni electrodes and GaSb NW. With a V_{DS} of 3V, the I_{DS} is 3.96 μ A. For the GaSb/GeS core-shell heterostructure NWs MSM photodetectors, the I_{DS} are significantly reduced compared to the pristine GaSb NW MSM photodetector. At the same time, the saturated I_{DS} are observed under large bias voltages. The saturated I_{DS} decreases from 304 to 217 and 12 nA for thin, medium, and thick shell NWs, respectively. In short, with the increase of shell thickness, the I_{DS} of as-studied MSM photodetectors decrease, indicating that the GaSb cores act as the main conductive channels (as shown in Fig. S7b), and the GeS shells limit the current.

To further illustrate the current limiting mechanism, a band structure model is depicted in Fig. S7c to demonstrate the hole transport process at the interface of GaSb and GeS. Under a forward bias voltage, the holes are injected from the drain electrode to the main conductive channel of the GaSb core and are collected at the source electrode, as shown in Fig. S7b. The holes pass across the GeS shell during both the injection and collection processes. As shown in Fig. S7c, a forward bias heterojunction and a reverse bias heterojunction are formed at the injection and collection regions, which block the hole transport and result in decreased current compared to the pristine GaSb NW MSM photodetector. At thermal equilibrium, the barrier of the forward bias heterojunction (due to the Fermi level coincident) is $q\psi_b$ (q is the electronic charge, and ψ_b is the electrostatic potential), and the barrier of the reverse bias heterojunction (due to the valence band offset) is ΔE_V (E_V is the valence band edge)^{5,6}. Obviously, with the increase of bias voltage, the barrier of the forward heterojunction of $q\psi_b$ at injection region decreases, which will benefit the hole transport, resulting in the increase of I_{DS} . At the same time, the barrier of the reverse heterojunction at the collection region keeps ΔE_V . In this case, the saturated I_{DS} is observed at a large bias voltage. It is worth mentioning that in addition to the classical thermionic emission, tunneling or defect-assisted tunneling also plays a role in the hole injection and collection processes^{7,8}. With a thin shell, the barrier is possibly thin

and low, benefiting the hole tunneling process. As a result, with the increase of shell thickness, the I_{DS} decreases." in the revised supplementary information.

Photoresponse at the contacts.

Since both the carrier injection and collection pass across the GeS shell, it also plays an important role in the transport of photogenerated carriers at the contacts. As discussed in the manuscript, the electron-hole pairs generated in the channel of core-shell heterostructure NWs will be collected at contacts under a bias voltage. The photogenerated holes also pass across the GeS shell at both the drain and source contacts. Obviously, the forward barrier of $q\psi_b$ dominates the transport of photogenerated holes (the reverse bias heterojunction is always ΔE_V). As shown in Fig. R2, the hole concentration of the GaSb core seriously affects the hole transport at the contacts (*Nat. Commun.* 2021, 12, 4094). When the hole concentration of the GaSb core decreases, the Fermi level of E_{F2} will rise (far away from the valence band), resulting in an increased hole transport barrier of $q\psi_b$ (Fig. R4b). The increased barrier will block the transport of holes, leading to a decreased current (negative photoresponse). On the contrary, when the hole concentration of the GaSb core increases, the E_{F2} will fall (close to the valence band), resulting in a decreased hole transport barrier (Fig. R4c). The decreased barrier will promote the transport of holes, leading to an increased current (positive photoresponse). Additionally, the thickness of the GeS shell also has an important impact on the hole transport in the photoresponse behavior, owing to the tunneling. From the dynamic photoresponse results in Fig. 4 and Fig. S15, with the shell thickness increasing from 11.3 ± 2.0 nm to 14.7 ± 1.9 nm and 19.5 ± 4.7 nm, the dark current decreases from 301 nA to 256 nA and 28 nA, and the photocurrent decreases from 73 nA to 64 nA and 6 nA under the illumination of 850 nm light, accordingly .

Fig. R2 | The photogenerated hole transport of GaSb/GeS core-shell heterostructure NW. a, Schematic of hole transport in the dark. **b-c,** Schematics of hole transport with the decreased and increased hole concentration in the GaSb core, respectively.

In this case, the discussion of "As a result, the injected electrons are going to recombine with the holes in the GaSb core (namely process III), resulting in the decreased hole concentration. On the other hand, the photogenerated electron-hole pairs in the GaSb core would increase the hole concentration. In fact, the hole

concentration of the GaSb core seriously affects the hole transport process at the contacts⁴⁷, as illustrated in Fig. S14. In a word, the decreased hole concentration in the GaSb core increases the hole transport barrier, leading to a decreased current (negative photoresponse). On the contrary, the increased hole concentration in the GaSb core decreases the hole transport barrier, leading to an increased current (positive photoresponse). Additionally, the thickness of the GeS shell also has an important impact on the hole transport in the photoresponse behavior, owing to the tunneling. The details can be found in the supplementary information." has been added on line 28 of page 12 in the revised manuscript.

At the same time, Fig. S14 and the discussion of "Since both the carrier injection and collection pass across the GeS shell, it also plays an important role in the transport of photogenerated carriers at the contacts. As discussed in the manuscript, the electron-hole pairs generated in the channel of core-shell heterostructure NWs will be collected at contacts under a bias voltage. The photogenerated holes also pass across the GeS shell at both the drain and source contacts. Obviously, the forward barrier of $q\psi_b$ dominates the transport of photogenerated holes (the reverse bias heterojunction is always ΔE_V). As shown in Fig. S14, the hole concentration of the GaSb core seriously affects the hole transport at the contact¹¹. When the hole concentration of the GaSb core decreases, the Fermi level of E_{F2} will rise (far away from the valence band), resulting in an increased hole transport barrier of $q\psi_b$ (Fig. S14b). The increased barrier will block the transport of holes, leading to a decreased current (negative photoresponse). On the contrary, when the hole concentration of the GaSb core increases, the E_{F2} will fall (close to the valence band), resulting in a decreased hole transport barrier (Fig. S14c). The decreased barrier will promote the transport of holes, leading to an increased current (positive photoresponse). Additionally, the thickness of the GeS shell also has an important impact on the hole transport in the photoresponse behavior, owing to the tunneling. From the dynamic photoresponse results in Fig. 4 and Fig. S15, with the shell thickness increasing from 11.3 ± 2.0 nm to 14.7 ± 1.9 nm and 19.5 ± 4.7 nm, the dark current decreases from 301 nA to 256 nA and 28 nA, and the photocurrent decreases from 73 nA to 64 nA and 6 nA under the illumination of 850 nm light, accordingly." have been added in the revised supplementary information.

Change List in the Revised Manuscript

1. On line 26 of page 10, "As shown in the I-V curves of Fig. S7, with the increase of shell thickness, the I_{DS} of as-studied MSM photodetectors decreases, indicating that the GaSb cores act as the main conductive channels, while the GeS shells limit the current owing to the formation of heterojunctions at the contacts. It is worth mentioning that in addition to the classical thermionic emission, tunneling or defect-assisted tunneling also plays a role in the hole injection and collection processes^{42,43}. With a thin shell, the barrier is possibly thin and low, benefiting the hole tunneling. As a result, with the increase in the shell thickness, the I_{DS} decreases. Since both the hole injection and collection processes pass across the GeS shell, it also plays an important role in the transport of photogenerated carriers at the contacts, and the details will be discussed later." has been added.
2. On line 28 of page 12, "As a result, the injected electrons are going to recombine with the holes in the GaSb core (namely process III), resulting in the decreased hole concentration. On the other hand, the photogenerated electron-hole pairs in the GaSb core would increase the hole concentration. In fact, the hole concentration of the GaSb core seriously affects the hole transport process at the contacts⁴⁷, as illustrated in Fig. S14. In a word, the decreased hole concentration in the GaSb core increases the hole transport barrier, leading to a decreased current (negative photoresponse). On the contrary, the increased hole concentration in the GaSb core decreases the hole transport barrier, leading to an increased current (positive photoresponse). Additionally, the thickness of the GeS shell also has an important impact on the hole transport in the photoresponse behavior, owing to the tunneling. The details can be found in the supplementary information." has been added.
3. On line 26 of page 13, "It is worth pointing out that the shell thickness also plays an important role in the visible light-assisted behavior, as shown in Fig. S15." has been revised.
4. On line 2 of page 15, "The significant improvement in the photodetection performance is attributed to the successful construction of the type-II p-n core-shell heterostructure⁵¹, as discussed in Fig. S16." has been revised.
5. On line 9 of page 15, "The ordered InGaAs/GeS core-shell NWs array is realized by a well-developed contact printing technology⁵², exhibiting excellent infrared detection capability with photocurrent up to 150 nA under the illumination of an 850 nm laser (Fig. S17)." has been revised.
6. On line 11 of page 16, "Furthermore, as shown in Fig. S18, the lattice-mismatch-free construction approach can also be used to grow chalcogenides core-shell heterostructure NWs." has been revised.

Other Change List in the Revised Manuscript

The section of "**References**" has been revised as below.

1. del Alamo, J. A. Nanometre-scale electronics with III-V compound semiconductors. *Nature* **479**, 317-323 (2011).
2. Kammhuber, J. *et al.* Conductance through a helical state in an Indium antimonide nanowire. *Nat. Commun.* **8**, 478 (2017).
3. Gibson, S. J. *et al.* Tapered InP nanowire arrays for efficient broadband high-speed single-photon detection. *Nat. Nanotechnol.* **14**, 473-479 (2019).
4. Mauthe, S. *et al.* High-speed III-V nanowire photodetector monolithically integrated on Si. *Nat. Commun.* **11**, 4565 (2020).
5. Peng, K. *et al.* Three-dimensional cross-nanowire networks recover full terahertz state. *Science* **368**, 510-513 (2020).
6. Nela, L. *et al.* Multi-channel nanowire devices for efficient power conversion. *Nat. Electron.* **4**, 284-290 (2021).
7. Yi, R. *et al.* Self-frequency-conversion nanowire lasers. *Light Sci. Appl.* **11**, 120 (2022).
8. Wang, J.-Y. *et al.* Supercurrent parity meter in a nanowire Cooper pair transistor. *Sci. Adv.* **8**, eabm9896 (2022).
9. Wang, D. *et al.* Observation of polarity-switchable photoconductivity in III-nitride/MoS_x core-shell nanowires. *Light Sci. Appl.* **11**, 227 (2022).
10. Mayer, B. *et al.* Lasing from individual GaAs-AlGaAs core-shell nanowires up to room temperature. *Nat. Commun.* **4**, 2931 (2013).
11. Balaghig, L. *et al.* Widely tunable GaAs bandgap via strain engineering in core/shell nanowires with large lattice mismatch. *Nat. Commun.* **10**, 2793 (2019).
12. Pendharkar, M. *et al.* Parity-preserving and magnetic field-resilient superconductivity in InSb nanowires with Sn shells. *Science* **372**, 508-511 (2021).
13. Oksenberg, E., Marti-Sanchez, S., Popovitz-Biro, R., Arbiol, J. & Joselevich, E. Surface-guided core-shell ZnSe@ZnTe nanowires as radial p-n heterojunctions with photovoltaic behavior. *ACS Nano* **11**, 6155-6166 (2017).
14. Wang, H. *et al.* Slowing hot-electron relaxation in mix-phase nanowires for hot-carrier photovoltaics. *Nano Lett.* **21**, 7761-7768 (2021).
15. Zhang, J. *et al.* Enhance the responsivity and response speed of self-powered ultraviolet photodetector by GaN/CsPbBr₃ core-shell nanowire heterojunction and hydrogel. *Nano Energy* **100**, 107437 (2022).
16. Treu, J. *et al.* Lattice-matched InGaAs-InAlAs core-shell nanowires with improved luminescence and photoresponse properties. *Nano Lett.* **15**, 3533-3540 (2015).
17. Lv, Q. *et al.* Lattice-mismatch-free growth of organic heterostructure nanowires from cocrystals to alloys. *Nat. Commun.* **13**, 3099 (2022).
18. Haapamaki, C. M., Baugh, J. & LaPierre, R. R. Critical shell thickness for InAs-Al_xIn_{1-x}As(P) core-shell nanowires. *J. Appl. Phys.* **112**, 124305 (2012).
19. Li, X. *et al.* Contactless optical characterization of carrier dynamics in free-standing InAs-InAlAs core-shell nanowires on silicon. *Nano Lett.* **19**, 990-996 (2019).
20. Arif, O. *et al.* Growth and strain relaxation mechanisms of InAs/InP/GaAsSb core-dual-shell nanowires. *Cryst. Growth Des.* **20**, 1088-1096 (2020).
21. Li, L. *et al.* Integrated flexible chalcogenide glass photonic devices. *Nat. Photonics* **8**, 643-649 (2014).
22. Lin, H. *et al.* Chalcogenide glass-on-graphene photonics. *Nat. Photonics* **11**, 798-805 (2017).
23. Jia, S. *et al.* Ultrahigh drive current and large selectivity in GeS selector. *Nat. Commun.* **11**, 4636 (2020).
24. Ohno, T. Passivation of GaAs(001) surfaces by chalcogen atoms (S, Se and Te). *Surf. Sci.* **255**, 229-236 (1991).
25. Sun, M. H. *et al.* Removal of surface states and recovery of band-edge emission in InAs nanowires through surface passivation. *Nano Lett.* **12**, 3378-3384 (2012).

26. Ohtake, A., Goto, S. & Nakamura, J. Atomic structure and passivated nature of the Se-treated GaAs(111) B surface. *Sci. Rep.* **8**, 1220 (2018).
27. Yang, Z.-x. *et al.* Surfactant-assisted chemical vapour deposition of high-performance small-diameter GaSb nanowires. *Nat. Commun.* **5**, 5249 (2014).
28. Sun, J. *et al.* Toward unusual-high hole mobility of p-channel field-effect-transistors. *Small* **17**, 2102323 (2021).
29. Ren, Z. *et al.* Short-wave near-infrared polarization sensitive photodetector based on GaSb nanowire. *IEEE Electron Device Lett.* **42**, 549-552 (2021).
30. Liu, D. *et al.* Schottky-contacted high-performance GaSb nanowires photodetectors enabled by lead-free all-inorganic perovskites decoration. *Small* **18**, 2200415 (2022).
31. Yang, Z.-x. *et al.* Complementary metal oxide semiconductor-compatible, high-mobility, < 111 >-oriented GaSb nanowires enabled by vapor-solid-solid chemical vapor deposition. *ACS Nano* **11**, 4237-4246 (2017).
32. Takebe, H., Maeda, H. & Morinaga, K. Compositional variation in the structure of Ge-S glasses. *J. Non-Cryst. Solids* **291**, 14-24 (2001).
33. Sakaguchi, Y., Hanashima, T., Ohara, K., Simon, A.-A. A. & Mitkova, M. Structural transformation in Ge_xS_{100-x} (10 ≤ x ≤ 40) network glasses: Structural varieties in short-range, medium-range, and nanoscopic scale. *Phys. Rev. Mater.* **3**, 035601 (2019).
34. Yin, Y. *et al.* Substrate-free chemical vapor deposition of large-scale III-V nanowires for high-performance transistors and broad-spectrum photodetectors. *Adv. Optical Mater.* **10**, 2102291 (2022).
35. Chen, H. *et al.* Termination of Ge surfaces with ultrathin GeS and GeS₂ layers via solid-state sulfurization. *Phys. Chem. Chem. Phys.* **19**, 32473-32480 (2017).
36. Zhao, S. *et al.* In situ growth of GeS nanowires with sulfur-rich shell for featured negative photoconductivity. *J. Phys. Chem. Lett.* **12**, 3046-3052 (2021).
37. Lauhon, L. J., Gudiksen, M. S., Wang, C. L. & Lieber, C. M. Epitaxial core-shell and core-multishell nanowire heterostructures. *Nature* **420**, 57-61 (2002).
38. Sa, Z. *et al.* Toward high bias-stress stability p-type GaSb nanowire field-effect-transistor for gate-controlled near-infrared photodetection and photocommunication. *Adv. Funct. Mater.* DOI: 10.1002/adfm.2023040 (2023).
39. Chen, Y. *et al.* Unipolar barrier photodetectors based on van der Waals heterostructures. *Nat. Electron.* **4**, 357-363 (2021).
40. Wang, D. *et al.* Bidirectional photocurrent in p-n heterojunction nanowires. *Nat. Electron.* **4**, 645-652 (2021).
41. Hwang, A. *et al.* Visible and infrared dual-band imaging via Ge/MoS₂ van der Waals heterostructure. *Sci. Adv.* **7**, eabj2521 (2021).
42. Ma, Q. *et al.* Tuning ultrafast electron thermalization pathways in a van der Waals heterostructure. *Nat. Phys.* **12**, 455-460 (2016).
43. Fan, S. *et al.* Tailoring quantum tunneling in a Vanadium-doped WSe₂/SnSe₂ heterostructure. *Adv. Sci.* **7**, 1902751 (2020).
44. Sun, J. *et al.* Stoichiometric effect on electrical and near-infrared photodetection properties of full-composition-range GaAs_{1-x}Sb_x nanowires. *Nano Res.* **14**, 3961-3968 (2021).
45. Yang, Y. *et al.* Hot carrier trapping induced negative photoconductance in InAs nanowires toward novel nonvolatile memory. *Nano Lett.* **15**, 5875-5882 (2015).
46. Fang, H. *et al.* Visible Light-assisted high-performance mid-infrared photodetectors based on single InAs nanowire. *Nano Lett.* **16**, 6416-6424 (2016).
47. Feng, S. *et al.* An ultrasensitive molybdenum-based double-heterojunction phototransistor. *Nat. Commun.* **12**, 4094 (2021).
48. Fang, H. & Hu, W. Photogating in low dimensional photodetectors. *Adv. Sci.* **4**, 1700323 (2017).
49. Bonilla, R. S. & Wilshaw, P. R. A technique for field effect surface passivation for silicon solar cells. *Appl. Phys. Lett.* **104**, 232903 (2014).
50. Mallorqui, A. D. *et al.* Field-effect passivation on silicon nanowire solar cells. *Nano Res.* **8**, 673-681

-
- (2015).
51. Zheng, Z., Zu, X., Zhang, Y. & Zhou, W. Rational design of type-II nano-heterojunctions for nanoscale optoelectronics. *Mater. Today Phys.* **15**, 100262 (2020).
 52. Fan, Z. *et al.* Wafer-scale assembly of highly ordered semiconductor nanowire arrays by contact printing. *Nano Lett.* **8**, 20-25 (2008).
 53. Dai, X. *et al.* GaAs/AlGaAs nanowire photodetector. *Nano Lett.* **14**, 2688-2693 (2014).
 54. Wang, X. *et al.* Vis-IR wide-spectrum photodetector at room temperature based on p-n junction-type GaAs_{1-x}Sb_x/InAs core-shell nanowire. *ACS Appl. Mater. Interfaces* **11**, 38973-38981 (2019).
 55. Li, H. *et al.* Novel type-II InAs/AlSb core-shell nanowires and their enhanced negative photocurrent for efficient photodetection. *Adv. Funct. Mater.* **28**, 1705382 (2018).
 56. Gallo, E. M. *et al.* Picosecond response times in GaAs/AlGaAs core/shell nanowire-based photodetectors. *Appl. Phys. Lett.* **98**, 241113 (2011).
 57. Zhou, C. *et al.* Self-assembly growth of In-rich InGaAs core-shell structured nanowires with remarkable near-infrared photoresponsivity. *Nano Lett.* **17**, 7824-7830 (2017).

Change List in the Revised Supplementary Information

1. Fig. S7, including the captions and corresponding discussion, has been revised as follows.

Fig. S7 | Electrical properties of GaSb/GeS core-shell heterostructure NWs. a, I-V curves of the pristine GaSb NW and GaSb/GeS core-shell heterostructure NWs with different shell thicknesses. **b,** Schematic of the NW MSM photodetector. **c,** Hole transport in the GaSb/GeS core-shell heterostructure NW.

Fig. S7 shows the electrical properties of as-constructed GaSb/GeS core-shell heterostructure NWs. Based on the I-V curves of Fig. S7a, the I_{DS} of the pristine GaSb NW MSM photodetector changes linearly with V_{DS} , demonstrating the typical ohmic-like contacts between Ni electrodes and GaSb NW. With a V_{DS} of 3V, the I_{DS} is 3.96 μ A. For the GaSb/GeS core-shell heterostructure NWs MSM photodetectors, the I_{DS} is significantly reduced compared to the pristine GaSb NW MSM photodetector. At the same time, the saturated I_{DS} is observed under large bias voltages. The saturated I_{DS} decreases from 304 to 217 and 12 nA for thin, medium, and thick shell NWs, respectively. In short, with the increase of shell thickness, the I_{DS} of as-studied MSM photodetectors decreases, indicating that the GaSb cores act as the main conductive channels (as shown in Fig. S7b), while the GeS shells limit the current.

To further illustrate the current limiting mechanism, a band structure model is depicted in Fig. S7c to demonstrate the hole transport process at the interface of GaSb and GeS. Under a forward bias voltage, the holes are injected from the drain electrode to the main conductive channel of GaSb core and are collected at the source electrode, as shown in Fig. S7b. The holes pass across the GeS shell during both the injection and collection processes. As shown in Fig. S7c, a forward bias heterojunction and a reverse bias heterojunction are formed at the injection and collection regions, which block the hole transport and result in decreased currents compared to the pristine GaSb NW MSM photodetector. At thermal equilibrium, the barrier of the forward bias heterojunction (due to the Fermi level coincident) is $q\psi_b$ (q is the electronic charge, and ψ_b is the electrostatic potential), and the barrier of the reverse bias heterojunction (due to the valence band offset) is ΔE_V (E_V is the valence band edge)^{5,6}. Obviously, with the increase of bias voltage, the barrier of the forward heterojunction of $q\psi_b$ at the injection region decreases, which will benefit the hole transport, resulting in the increase of I_{DS} . At the same time, the barrier of the reverse heterojunction at the collection region keeps ΔE_V . In this case, the saturated I_{DS} is observed at a large bias voltage. It is worth

mentioning that in addition to the classical thermionic emission, tunneling or defect-assisted tunneling also plays a role in the hole injection and collection processes^{7,8}. With a thin shell, the barrier is possibly thin and low, benefiting the hole tunneling process. As a result, with the increase in shell thickness, the I_{DS} decreases.

2. Fig. S14, including the captions and corresponding discussion, has been added as follows.

Fig. S14 | The photogenerated hole transport of the GaSb/GeS core-shell heterostructure NW. a, Schematic of hole transport in the dark. **b-c,** Schematics of hole transport with the decreased and increased hole concentration in GaSb core, respectively.

Since both the carrier injection and collection pass across the GeS shell, it also plays an important role in the transport of photogenerated carriers at the contacts. As discussed in the manuscript, the electron-hole pairs generated in the channel of core-shell heterostructure NWs will be collected at contacts under a bias voltage. The photogenerated holes also pass across the GeS shell at both the drain and source contacts. Obviously, the forward barrier of $q\psi_b$ dominates the transport of photogenerated holes (the reverse bias heterojunction is always ΔE_v). As shown in Fig. S14, the hole concentration of the GaSb core seriously affects the hole transport at the contacts¹¹. When the hole concentration of the GaSb core decreases, the Fermi level of E_{F2} will rise (far away from the valence band), resulting in an increased hole transport barrier of $q\psi_b$ (Fig. S14b). The increased barrier will block the transport of holes, leading to a decreased current (negative photoresponse). On the contrary, when the hole concentration of the GaSb core increases, the E_{F2} will fall (close to the valence band), resulting in a decreased hole transport barrier (Fig. S14c). The decreased barrier will promote the transport of holes, leading to an increased current (positive photoresponse). Additionally, the thickness of the GeS shell also has an important impact on the hole transport in the photoresponse behavior, owing to the tunneling. From the dynamic photoresponse results in Fig. 4 and Fig. S15, with the shell thickness increasing from 11.3 ± 2.0 nm to 14.7 ± 1.9 nm and 19.5 ± 4.7 nm, the dark current decreases from 301 nA to 256 nA and 28 nA, and the photocurrent decreases from 73 nA to 64 nA and 6 nA under the illumination of 850 nm light, accordingly.

3. The section of "References" has been revised as below.

1. Gobeli, G. W. & Allen, F. G. Photoelectric properties of cleaved GaAs GaSb InAs and InSb surfaces-comparison with Si and Ge. *Phys. Rev.* **137**, A245-A254 (1965).
2. Jia, S. *et al.* Ultrahigh drive current and large selectivity in GeS selector. *Nat. Commun.* **11**, 4636 (2020).
3. Zhang, D. & Li, Z. InP/ZnS quantum dots functionalized AlGaAs/InGaAs open gate high electron mobility transistor. *J. Mater. Sci.: Mater. Electron.* **29**, 10663-10668, (2018).
4. Ahn, H.-W. *et al.* Effect of density of localized states on the ovonic threshold switching characteristics of the amorphous GeSe films. *Appl. Phys. Lett.* **103**, 042908 (2013).
5. Hu, W. *et al.* Germanium/perovskite heterostructure for high-performance and broadband photodetector from visible to infrared telecommunication band. *Light Sci. Appl.* **8**, 106 (2019).
6. Wu, F. *et al.* High efficiency and fast van der Waals hetero-photodiodes with a unilateral depletion region. *Nat. Commun.* **10**, 4663 (2019).
7. Ma, Q. *et al.* Tuning ultrafast electron thermalization pathways in a van der Waals heterostructure. *Nat. Phys.* **12**, 455-460 (2016).
8. Fan, S. *et al.* Tailoring quantum tunneling in a Vanadium-doped WSe₂/SnSe₂ heterostructure. *Adv. Sci.* **7**, 1902751 (2020).
9. Lee, H. K. H. *et al.* The role of fullerenes in the environmental stability of polymer: fullerene solar cells. *Energy Environ. Sci.* **11**, 417-428 (2018).
10. Caprioglio, P. *et al.* On the origin of the ideality factor in perovskite solar cells. *Adv. Energy Mater.* **10**, 2000502 (2020).
11. Feng, S. *et al.* An ultrasensitive molybdenum-based double-heterojunction phototransistor. *Nat. Commun.* **12**, 4094 (2021).
12. Bonilla, R. S. & Wilshaw, P. R. A technique for field effect surface passivation for silicon solar cells. *Appl. Phys. Lett.* **104**, 232903 (2014).
13. Mallorqui, A. D. *et al.* Field-effect passivation on silicon nanowire solar cells. *Nano Res.* **8**, 673-681 (2015).
14. Lu, Z. *et al.* Ultrahigh speed and broadband few-layer MoTe₂/Si 2D-3D heterojunction-based photodiodes fabricated by pulsed laser deposition. *Adv. Funct. Mater.* **30**, 1907951 (2020).
15. Dai, X. *et al.* GaAs/AlGaAs nanowire photodetector. *Nano Lett.* **14**, 2688-2693 (2014).
16. Wang, X. *et al.* Vis-IR wide-spectrum photodetector at room temperature based on p-n junction-type GaAs_{1-x}Sb_x/InAs core-shell nanowire. *ACS Appl. Mater. Interfaces* **11**, 38973-38981 (2019).
17. Li, H. *et al.* Novel type-II InAs/AlSb core-shell nanowires and their enhanced negative photocurrent for efficient photodetection. *Adv. Funct. Mater.* **28**, 1705382 (2018).
18. Gallo, E. M. *et al.* Picosecond response times in GaAs/AlGaAs core/shell nanowire-based photodetectors. *Appl. Phys. Lett.* **98**, 241113 (2011).
19. Zhou, C. *et al.* Self-assembly growth of In-rich InGaAs core-shell structured nanowires with remarkable near-infrared photoresponsivity. *Nano Lett.* **17**, 7824-7830 (2017).